# UNDERSTANDING DOMAIN GENERALIZATION: A VIEW OF NECESSITY AND SUFFICIENCY

## ABSTRACT

Despite the rapid advancements in domain generalization (DG), the majority of DG studies center on establishing theoretical guarantee for generalization under the assumption of sufficient, diverse or even infinite domains. This assumption however is unrealistic, thus there remains no conclusive evidence as to whether the existing DG algorithms can truly generalize in practical settings where domains are limited. This paper aims to elucidate this matter. We first study the conditions for the existence and learnability of an optimal hypothesis. As the sufficient conditions are non-verifiable, our identified two necessary conditions become critical to guaranteeing the chance of finding the global optimal hypothesis in finite domain settings. In light of the theoretical insights, we provide a comprehensive review of DG algorithms explaining to what extent they can generalize effectively. We finally introduce a practical approach that leverages the joint effect of the two sets of conditions to boost generalization. Our proposed method demonstrates superior performance on well-established DG benchmarks.

## 1 INTRODUCTION

Domain generalization (DG) aims to train a machine learning model on multiple data distributions so that it can generalize to unseen data distributions. Although challenging, DG is crucial for practical scenarios where there is a need to quickly deploy a prediction model on a new target domain without access to target data. Various approaches have been proposed to address the DG problem, which can be broadly categorized into 3 families: representation alignment, invariant prediction, and data augmentation. *Representation alignment* focuses on learning domain-invariant representations by reducing the divergence between latent marginal distributions (Long et al., 2017; Ganin et al., 2016; Li et al., 2018b; Nguyen et al., 2021; Shen et al., 2018; Xie et al., 2017; Ilse et al., 2020) or aligning conditional distributions (Gong et al., 2016; Li et al., 2018d; Tachet des Combes et al., 2020). *Invariant prediction* ensures stable performance regardless of the domain by learning a consistently optimal classifier (Arjovsky et al., 2020; Ahuja et al., 2020; Krueger et al., 2021; Rosenfeld et al., 2020; Li et al., 2022a). *Data augmentation* applies predefined or learnable transformations on the original samples or their features to create augmented data, thereby enhancing the model's generalization capabilities (Mitrovic et al., 2020; Wang et al., 2022b; Shankar et al., 2018; Zhou et al., 2020; 2021; Xu et al., 2021; Zhang et al., 2017; Wang et al., 2020b; Zhao et al., 2020; Yao et al., 2022a; Carlucci et al., 2019; Yao et al., 2022b). Despite these developments, these methods have not consistently outperformed Empirical Risk Minimization (ERM) on fair model selection criteria (Gulrajani & Lopez-Paz, 2021; Idrissi et al., 2022; Ye et al., 2022; Chen et al., 2022a).

Several studies have sought to elucidate this phenomenon. In one line of research, the prevailing theoretical models in DG are typically established based on *domain adaptation* (Ben-David et al., 2010; Ben-Hur et al., 2001; Phung et al., 2021; Zhou et al., 2020; Johansson et al., 2019), which mainly discuss the differences between source and target domains. In other approaches grounded in causality, namely (Arjovsky et al., 2020; Mitrovic et al., 2020; Zhang et al., 2023), there is typically an assumption of having prior knowledge of target domains. There are also studies on the optimality conditions for generalization. Specifically, Ruan et al. (2021)[1] require the optimal representation to be discriminative for the task and the representation's marginal support to be same across source and target, which theoretically, (Ruan et al., 2021) also require knowledge of target domains.

---

[1]We further elaborate on the connection between (Ruan et al., 2021) and our work in Appendix B.

Table 1: Summary of Conditions for Generalization

| Condition | Type | Target DG approach |
|---|---|---|
| Label-identifiability (3.1) | Assumption | |
| Causal support (3.2) | Assumption | |
| Optimal hypothesis for $\mathcal{E}_{tr}$ + Sufficient and diverse domains (3.4) | Sufficient | Data augmentation |
| Optimal hypothesis for $\mathcal{E}_{tr}$ + Invariant representation function (3.5) | Sufficient | Representation alignment & Invariant prediction |
| Optimal hypothesis for $\mathcal{E}_{tr}$ (3.3) | Necessary | |
| **Sufficient Representation Function** (3.7) | **Necessary** | Ensembles |

While these analyses offer insights about DG from various perspectives, we argue that these conclusions do not fully contribute to our understanding of generalization in practice where only a *finite* number of training domains are available. Indeed, these frameworks establish generalization either under the condition that target domains are known or diverse, or when sufficient number of training domains are given. Consequently, it remains largely unknown regarding the extent to which domain generalization can be attained in limited and finite number of domains, as well as the nature of the representation required to achieve this. Our work seeks to fill in this gap with a comprehensive study of DG landscape in light of the following aspects:

**1. Conditions for Generalization.** We first systematically develop a set of necessary and sufficient conditions for generalization. We reaffirm that although existing DG methods strive to achieve the sufficient conditions, these conditions remain non-verifiable, thus cannot guarantee the chance of reaching a global optimal hypothesis when training domains are only finite (See Section 3).

**2. DG through the lens of Necessity and Sufficiency.** We then shed light on how the DG dynamics is greatly reshaped in limited domain settings. Our analysis reveals that when a sufficient condition (3.5 or 3.4 in Table 1) is met, it automatically results in the fulfillment of both necessary conditions (3.3 and 3.7 in Table 1). DG literature thus tends to overlook the role of the necessary conditions in real-world scenarios, particularly the condition of *sufficient representation function* (3.7). When the sufficient conditions cannot be guaranteed, the necessary conditions in fact hold greater practical value in determining how to maximize the likelihood of achieving generalization (See Section 4.1). This licenses a new view to understanding why DG algorithms fail to outperform the fundamental approach of empirical risk minimization (ERM) on standard benchmarks (See Section 4.2).

**3. Learning Sufficient Invariant Representation.** Finally, we empirically validate our theories by proposing a practical method that promotes the *sufficient representation* constraint via ensemble learning, while maintains the necessary conditions via a novel representation alignment strategy. Our method demonstrates superior performance across all experimental settings (See Section 5).

## 2 PRELIMINARIES

We first introduce the notations and basic concepts in the paper. We use calligraphic letters (i.e., $\mathcal{X}$) for spaces, upper case letters (i.e. $X$) for random variables, lower case letters (i.e. $x$) for their values and $\mathbb{P}$ for (observed) probability distributions.

### 2.1 PROBLEM SETUP

We consider a standard domain generalization setting with a potentially high-dimensional variable $X$ (e.g., an image), a label variable $Y$ and a discrete environment (or domain) variable $E$ in the sample spaces $\mathcal{X}, \mathcal{Y}$, and $\mathcal{E}$, respectively. We consider the following family of distributions over the observed variables $(X, Y)$ given the environment $E = e \in \mathcal{E}$ where environment space under consideration $\mathcal{E} = \{e \mid \mathbb{P}^e \in \mathcal{P}\}$:

$$\mathcal{P} = \left\{ \mathbb{P}^e(X, Y) = \int_{z_c} \int_{z_e} \mathbb{P}(X, Y, Z_c, Z_e, E = e) dz_c dz_e \right\}$$

The data generative process underlying every observed distribution $\mathbb{P}^e(X, Y)$ is characterized by a *structural causal model* (SCM) over a tuple $\langle V, U, \psi \rangle$ (See Figure 1). The SCM consists of a set of *endogenous* variables $V = \{X, Y, Z_c, Z_e, E\}$, a set of mutually independent *exogenous* variables $U = \{U_x, U_y, U_{z_c}, U_{z_e}, U_e\}$ associated with each variable in $V$ and a set of deterministic equations

$\psi = \{\psi_x, \psi_y, \psi_{z_c}, \psi_{z_e}, \psi_e\}$ representing the generative process for $V$. We note that this generative structure has been widely used and extended in several other studies, including (Chang et al., 2020; Mahajan et al., 2021; Li et al., 2022a; Zhang et al., 2023; Lu et al., 2021; Liu et al., 2021).

The generative process begins with the sampling of an environmental variable $e$ from a prior distribution $\mathbb{P}(U_e)$[2]. We assume there exists a causal factor $z_c \in \mathcal{Z}_c$ determining the label $Y$ and a environmental feature $z_e \in \mathcal{Z}_e$ *spuriously* correlated with $Y$. These two latent factors are generated from an environment $e$ via the mechanisms $z_c = \psi_{z_c}(e, u_{z_c})$ and $z_e = \psi_{z_e}(e, u_{z_e})$ with $u_{z_c} \sim \mathbb{P}(U_{z_c}), u_{z_e} \sim \mathbb{P}(U_{z_e})$. A data sample $x \in \mathcal{X}$ is generated from both the causal feature and the environmental feature i.e., $x = \psi_x(z_c, z_e, u_x)$ with $u_x \sim \mathbb{P}(U_x)$.

Figure 1 dictates that the joint distribution over $X$ and $Y$ can vary across domains resulting from the variations in the distributions of $Z_c$ and $Z_e$. Furthermore, *both causal and environmental features are correlated with $Y$, but only $Z_c$ causally influences $Y$*. However because $Y \perp\!\!\!\perp E | Z_c$, the conditional distribution of $Y$ given a specific $Z_c = z_c$ remains unchanged across different domains i.e., $\mathbb{P}^e(Y|Z_c = z_c) = \mathbb{P}^{e'}(Y|Z_c = z_c) \ \forall e, e' \in \mathcal{E}$. For readability, we omit the superscript $e$ and denote this invariant conditional distribution as $\mathbb{P}(Y|Z_c = z_c)$.

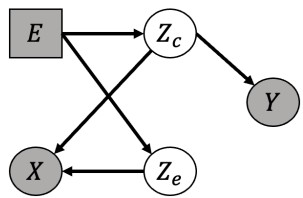

Figure 1: A directed acyclic graph (DAG) describing the causal relations among different factors producing data $X$ and label $Y$ in our SCM. Observed variables are shaded.

## 2.2 REVISITING DOMAIN GENERALIZATION SETTING

*Domain objective*: Given a domain $\mathbb{P}^e$, let the hypothesis $f : \mathcal{X} \to \Delta_{|\mathcal{Y}|}$ is a map from the data space $\mathcal{X}$ to the the $C$-simplex label space $\Delta_{|\mathcal{Y}|} := \{\alpha \in \mathbb{R}^{|\mathcal{Y}|} : \|\alpha\|_1 = 1 \wedge \alpha \geq 0\}$. Let $l : \mathcal{Y}_\Delta \times \mathcal{Y} \mapsto \mathbb{R}$ be a loss function, where $\ell(f(x), y)$ with $f(x) \in \mathcal{Y}_\Delta$ and $y \in \mathcal{Y}$ specifies the loss (i.e., cross-entropy) to assign a data sample $x$ to the class $y$ by the hypothesis $f$. The general loss of the hypothesis $f$ w.r.t. a given domain $\mathbb{P}^e$ is:

$$\mathcal{L}(f, \mathbb{P}^e) := \mathbb{E}_{(x,y) \sim \mathbb{P}^e}[\ell(f(x), y)]. \tag{1}$$

*Domain Generalization*: Given a set of training domains $\mathcal{E}_{tr} = \{e_1, ..., e_K\} \subset \mathcal{E}$, the objective of DG is to exploit the 'commonalities' present in the training domains to improve generalization to any domain of the population $e \in \mathcal{E}$. For supervised classification, the task is equivalent to seeking the set of **global optimal hypotheses** $\mathcal{F}^*$ where every $f \in \mathcal{F}^*$ is locally optimal for every domain:

$$\mathcal{F}^* := \bigcap_{e \in \mathcal{E}} \operatorname*{argmin}_{f \in \mathcal{F}} \mathcal{L}(f, \mathbb{P}^e) \tag{2}$$

We here examine the widely used *composite hypothesis* $f = h \circ g \in \mathcal{F}$, where $g : \mathcal{X} \to \mathcal{Z}$ belongs to a set of representation functions $\mathcal{G}$, mapping the data space $\mathcal{X}$ to a latent space $\mathcal{Z}$, and $h : \mathcal{Z} \to \Delta_{|\mathcal{Y}|}$ is the classifier in the space $\mathcal{H}$. For simplicity, we assume $\mathcal{Z}_c, \mathcal{Z}_e \subseteq \mathcal{Z}$ in the following analyses.

**Presumption**. While our work considers limited and finite domains, we follow recent theoretical works (Wang et al., 2022a; Rosenfeld et al., 2020; Kamath et al., 2021; Ahuja et al., 2021; Chen et al., 2022b) assuming the infinite data setting for every training environment. This assumption distinguishes DG literature from traditional generalization analysis (e.g., PAC-Bayes framework) that focuses on in-distribution generalization where the testing data are drawn from the same distribution.

## 3 CONDITIONS FOR GENERALIZATION

In this section, we present the key assumptions about the data setting along with the necessary and sufficient conditions on the hypothesis and representation functions for achieving generalization

---

[2]explicitly via the equation $e = \psi_e(u_e), u_e \sim P(U_e)$.

defined in Eq. (2) (See Table 1 for summary). These conditions are critical to our analysis, where we first reveal that the existing DG methods aim to satisfy one or several of these necessary and sufficient conditions to achieve generalization. We thus thereafter theoretically assess whether a method works effectively by to what extent the necessary conditions are met.

## 3.1 Assumptions on Data Setting

We first establish crucial assumptions for the feasibility of generalization as described in Eq (2). These assumptions are essential for understanding the conditions under which generalization can be achieved. We also demonstrate that the first assumption is a necessary condition for the existence of global optimal hypotheses (Appendix A.3).

**Assumption 3.1.** (Label-identifiability). We assume that for any pair $z_c, z_c^{'} \in \mathcal{Z}_c$, $\mathbb{P}(Y|Z_c = z_c) = \mathbb{P}(Y|Z_c = z_c^{'})$ if $\psi_x(z_c, z_e, u_x) = \psi_x(z_c', z_e', u_x')$ for some $z_e, z_e', u_x, u_x'$.

The causal graph indicates that $Y$ is influenced by $z_c$, making $Y$ identifiable over the distribution $\mathbb{P}(Z_c)$. This assumption implies that different causal factors $z_c$ and $z_c^{'}$ cannot yield the same $x$, unless the condition $\mathbb{P}(Y|Z_c = z_c) = \mathbb{P}(Y|Z_c = z_c^{'})$ holds, or the distribution $\mathbb{P}(Y \mid x)$ is stable. This assumption also can be view as covariate shift setting in OOD (Shimodaira, 2000).

**Assumption 3.2.** (Causal support). We assume that the union of the support of causal factors across training domains covers the entire causal factor space $\mathcal{Z}_c$: $\cup_{e \in \mathcal{E}_{tr}} \text{supp}\{\mathbb{P}^e(Z_c)\} = \mathcal{Z}_c$ where $\text{supp}(\cdot)$ specifies the support set of a distribution.

This assumption holds significance in DG theories (Johansson et al., 2019; Ruan et al., 2021; Li et al., 2022b), especially when we avoid imposing strict constraints on the target functions. Particularly, (Ahuja et al., 2021) showed that without the support overlap assumption on the causal features, OOD generalization is impossible for such a simple model as linear classification. Meanwhile, for more complicated tasks, deep neural networks are typically employed, which, when trained via gradient descent however, cannot effectively approximate a broad spectrum of nonlinear functions beyond their support range (Xu et al., 2020). It is worth noting that causal support overlap does not imply that the distribution over the causal features is held unchanged.

## 3.2 Conditions on Hypothesis

By definition, the global optimal hypothesis $f \in \mathcal{F}^*$ must also be the optimal solution for all training domains in $\mathcal{E}_{tr}$ which is defined as

**Definition 3.3.** (Optimal hypothesis for training domains) Given $\mathcal{F}_{\mathbb{P}^e} = \underset{f \in \mathcal{F}}{argmin} \ \mathcal{L}(f, \mathbb{P}^e)$ is set of optimal hypothesis for $\mathbb{P}^e$, the optimal hypothesis for all training domains $f \in \mathcal{F}_{\mathcal{E}_{tr}} = \bigcap_{e \in \mathcal{E}_{tr}} \mathcal{F}_{\mathbb{P}^e}$.

It is evident that a hypothesis being optimal for all training domains is a *necessary condition* for achieving a global optimal hypothesis (if $f \in \mathcal{F}^*$ then $f \in \mathcal{F}_{\mathcal{E}_{tr}}$). Since this condition is necessary, the reverse does not hold i.e., $f \in \mathcal{F}_{\mathcal{E}_{tr}}$ does not guarantee $f \in \mathcal{F}^*$. However, this condition remains essential for our theoretical analysis as it is the only condition that can be verified during training. We next present two sufficient conditions that lay the foundation for understanding DG algorithms.

## 3.3 Conditions on Training Domains

We are thus motivated to study the properties of the training domains $\mathcal{E}_{tr}$ so that it is feasible to capture the global optimal hypothesis from these domains.

**Theorem 3.4.** *(Sufficient and diverse domains) Given sequence of training domains $\mathcal{E}_{tr} = \{e_1, ..., e_K\} \subset \mathcal{E}$, denote $\mathcal{F}_\cap^k = \bigcap_{i=1}^k \mathcal{F}_{\mathbb{P}^{e_i}}$. We consider $\mathcal{E}_{tr}$ to be **diverse** if for domain $e_k$, there exists at least one sample $x = \psi_x(z_c, z_e, u_x) \in supp\{\mathbb{P}^{e_k}(X)\}$ such that $\exists f \in \mathcal{F}_\cap^{k-1} : f(x) \neq \mathbb{P}(Y \mid z_c)$. Given a set of diverse domains $\mathcal{E}_{tr}$, we have:*

$$\mathcal{F}_\cap^1 \supset \mathcal{F}_\cap^2 \supset ... \supset \mathcal{F}_\cap^K$$

*and the number of training domains $\mathcal{E}_{tr}$ is sufficiently large:*

$$\lim_{\mathcal{E}_{tr} \to \mathcal{E}} \mathcal{F}_\cap^{|\mathcal{E}_{tr}|} \to \mathcal{F}^*.$$

*(Proof in Appendix A.9)*

Theorem 3.4 dictates that having a sufficiently large and diverse set of training domains is a *sufficient condition* for attaining the global optimal hypothesis. However, our theorem does not explicitly specify *how large* the number of training domains must be. For a more in-depth study on this aspect, we refer readers to (Rosenfeld et al., 2020; Arjovsky et al., 2020). In this work, we focus on the "diversity" property since it is generally difficult to determine how many domains is enough but we can always attempt to make them diverse, as done by the family of augmentation-based DG algorithms (refer to Section 4.2). These algorithms, such as (Mitrovic et al., 2020; Wang et al., 2022b), create augmented data that preserve the causal factor $z_c$ while varying the environment factor $z_e$ to encourage the classifier to focus on exploiting the causal factor $z_c$. It is worth noting that while these methods aim to achieve *sufficient condition* 3.4, the condition is, in fact, theoretically non-verifiable without knowledge of the target domains.

### 3.4 CONDITIONS ON REPRESENTATION FUNCTION

**Proposition 3.5.** *(Invariant Representation Function) Under Assumption.3.1, there exists a set of deterministic representation function $(\mathcal{G}_c \neq \emptyset) \in \mathcal{G}$ such that for any $g \in \mathcal{G}_c$, $\mathbb{P}(Y \mid g(x)) = \mathbb{P}(Y \mid z_c)$ and $g(x) = g(x')$ holds true for all $\{(x, x', z_c) \mid x = \psi_x(z_c, z_e, u_x), x' = \psi_x(z_c, z_e', u_x')$ for all $z_e, z_e', u_x, u_x'\}$ (Proof in Appendix A.4).*

Assumption 3.1 gives rise to a family of invariant representation function $\mathcal{G}_c$, as stated in Proposition 3.5. This discovery points to the presence of global optimal hypotheses i.e., $\mathcal{F}^* \neq \emptyset$. Furthermore, in the subsequent theorem, we demonstrate that with an understanding of the invariant correlation $g \in \mathcal{G}_c$, it is possible to learn these global optimal hypotheses from any training dataset $\mathbb{P}^e \sim \mathcal{P}$, given it exhibits *sufficient causal support* (e.g., a mixture of training domains under Assumption 3.2, where $\cup_{e \in \mathcal{E}_{tr}} \text{supp}\{\mathbb{P}^e(Z_c)\} = \mathcal{Z}_c$).

**Theorem 3.6.** *Denote the set of **domain optimal hypotheses** of $\mathbb{P}^e$ induced by $g \in \mathcal{G}$:*

$$\mathcal{F}_{\mathbb{P}^e, g} = \left\{ h \circ g \mid h \in \underset{h' \in \mathcal{H}}{\operatorname{argmin}} \mathcal{L}\left(h' \circ g, \mathbb{P}^e\right) \right\}.$$

*If $supp\{\mathbb{P}^e(Z_c)\} = \mathcal{Z}_c$ and $g \in \mathcal{G}_c$, then $\mathcal{F}_{\mathbb{P}^e, g} \subseteq \mathcal{F}^*$. (Proof in Appendix A.6)*

Theorem 3.6 demonstrate that under Assumption 3.1 and Assumption 3.2, $g \in \mathcal{G}_c$ is the *sufficient condition $f^* \in \mathcal{F}^*$* for learning global optimal hypothesis from finite number of training domains. This condition is what the family of *representation alignment* and *invariant prediction* methods strives at (refer to Section 4.2). However, achieving $g \in \mathcal{G}_c$ is often infeasible in practice, because it requires the knowledge of *all* domains. We therefore shift the attention to studying a new class of representation function that serves as a *necessary condition* for global optimal hypothesis, which is defined as follows:

**Definition 3.7.** *(Sufficient Representation Function) A set of representation functions $\mathcal{G}_s \in \mathcal{G}$ is considered as sufficient representation functions if for any $g \in \mathcal{G}_s$, there exists a function $\phi : \mathcal{Z} \to \mathcal{Z}$ such that $(\phi \circ g) \in \mathcal{G}_c$ (i.e., given $g \in \mathcal{G}_s$, $g(x)$ retains all information about causal feature of $x$).*

The following theorem shows that $g \in \mathcal{G}_s$ is necessary for achieving the global optimal hypothesis.

**Theorem 3.8.** *Considering the training domains $\mathbb{P}^e$ and representation function $g$, let $\mathcal{H}_{\mathbb{P}^e, g} = argmin_{h \in \mathcal{H}} \mathcal{L}\left(h \circ g, \mathbb{P}^e\right)$ represent the set of optimal classifiers on $g\#\mathbb{P}^e$ (the push-forward distribution by applying $g$ on $\mathbb{P}^e$), **the best generalization classifier** from $\mathbb{P}^e$ to $\mathcal{P}$ is defined as*

$$\mathcal{F}_{\mathbb{P}^e, g}^B = \left\{ h \circ g \mid h \in \bigcap_{e' \in \mathcal{E}} \underset{h' \in \mathcal{H}_{\mathbb{P}^e, g}}{\operatorname{argmin}} \mathcal{L}\left(h' \circ g, \mathbb{P}^{e'}\right) \right\} \quad (3)$$

*Give representation function $g : \mathcal{X} \to \mathcal{Z}$ then $\forall \mathbb{P}^e \sim \mathcal{P}$ we have $\left(\mathcal{F}_{\mathbb{P}^e, g}^B \neq \emptyset\right) \subseteq \mathcal{F}^*$ if and only if $g \in \mathcal{G}_s$. (Proof in Appendix A.7)*

This theorem demonstrates that if $g$ is not a sufficient representation i.e., $g \notin \mathcal{G}_s$, the best attainable hypothesis is surely not optimal i.e., $\mathcal{F}_{\mathbb{P}^e, g}^B \cap \mathcal{F}^* = \emptyset$, implying that it is impossible to find any classifier $h$ such that $h \circ g \in \mathcal{F}^*$. In other words, $g \in \mathcal{G}_s$ is *necessary* for $f \in \mathcal{F}^*$. This property plays a crucial role in understanding the generalization ability of DG algorithms.

# 4 DOMAIN GENERALIZATION: A VIEW OF NECESSITY AND SUFFICIENCY

## 4.1 CAN DG ALGORITHMS GENERALIZE?

The majority of DG algorithms strive to satisfy one of the sufficient conditions to achieve generalization. However, the sufficient conditions are nearly non-verifiable when training domains are limited. Corollary 4.1 indicates that if a hypothesis $f = h \circ g$ satisfies all necessary conditions but fails to meet any sufficient condition, it may still perform poorly in many target domains. That is, there might exist $f \in \bigcap_{e \in \mathcal{E}_{tr}} \mathcal{F}_{g,\mathbb{P}^e}$ but $f \notin \mathcal{F}^*$ and if $f \notin \mathcal{F}^*$, there are many "bad" domains $P^T$ for which loss $\mathcal{L}(f, P^T)$ is arbitrary large (recall that $\mathcal{F}^*$ is set of globally optimal hypotheses).

**Corollary 4.1.** *Given $g \in \mathcal{G}_s$, there exists $f = h \circ g \in \bigcap_{e \in \mathcal{E}_{tr}} \mathcal{F}_{g,\mathbb{P}^e}$ such that for any $0 \leq \delta \leq 1$, there are many undesirable target domains $\mathbb{P}^T \sim \mathcal{P}$ such that:*

$$\mathbb{E}_{(x,y)\sim\mathbb{P}^T}\left[f(x) \neq f^*(x)\right] \geq 1 - \delta.$$

*with $f^* \in \mathcal{F}^*$.[3]  (Proof in Appendix A.8)*

A natural question is to what extent DG algorithms are generalizable when the sufficient conditions cannot be guaranteed. In this case, generalizability depends on how well they can address the necessary conditions. If a DG algorithm violates our necessary conditions, the chance of achieving generalizing is in fact zero. Without considering these conditions, it remains undetermined whether the algorithm can ever reach a global optimal hypothesis.

Let us denote $\mathcal{F}_{\mathbb{P}^e} = \bigcup_{g \in \mathcal{G}_s} \mathcal{F}_{\mathbb{P}^e,g}$ as the set of hypotheses induced by an algorithm $A$ that satisfies both necessary conditions i.e., optimal for domain $e$ (*Condition* 3.3) and $g \in \mathcal{G}_s$ is a *sufficient representation* (*Condition* 3.7). Given that a hypothesis $f \in \mathcal{F}^*$ must also be the optimal solution for all training domains in $\mathcal{E}_{tr}$, we deduce that $\mathcal{F}^* \subseteq \mathcal{F}_{\mathbb{P}^e}, \forall e \in \mathcal{E}_{train}$, consequently, $\mathcal{F}^* \subseteq \bigcap_{e \in \mathcal{E}_{train}} \mathcal{F}_{\mathbb{P}^e}$. This relationship is illustrated in the Venn diagrams of Figure 2.

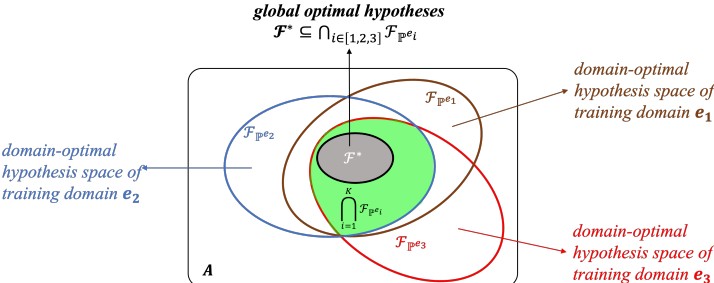

Figure 2: The circles (brown, blue, red) denote the spaces of domain-optimal hypotheses $\mathcal{F}_{\mathbb{P}^{e_1}}$, $\mathcal{F}_{\mathbb{P}^{e_2}}$, $\mathcal{F}_{\mathbb{P}^{e_3}}$ of training domains $e_1, e_2, e_3$ respectively. The grey area indicates the space of global optimal hypotheses $\mathcal{F}^*$. An algorithm $A$ satisfying both conditions 3.3 and 3.7 induces a non-empty grey area that lies within the green area - the joint space of domain-optimal hypotheses $\bigcap_{i \in \{1,2,3\}} \mathcal{F}_{\mathbb{P}^{e_i}}$.

The Venn diagram reveals that any algorithm achieving *Condition* 3.3 can guarantee that its corresponding global optimal set is bounded by the feasible hypothesis set induced by the algorithm; in visual terms, the green area always cover the grey area. Apparently, "bad" domains occur for any learned hypothesis $f$ that falls outside of the grey area. Therefore, the more the green area collapses to the grey area, the higher chance generalization can be attained.

At a high level, a strategy to encourage both areas coincide is thus by reducing the size of the green area with additional constraints that can be met by all hypotheses in the grey area. Especially when either of the sufficient conditions is met, according to Theorem 3.4, the green area can always converge to the grey area under perfect optimization. Existing DG approaches are essentially seeking to reduce the green area. The *augmentation*-based methods strive to achieve *sufficient and diverse domains* (a sufficient condition) by generating augmented domains to challenge the hypothesis. Meanwhile, *Representation alignment* or *invariant prediction* strategies implicitly narrow down the green space by constraining the representation function space $\mathcal{G}$.

---

[3]This coincides with the "no free lunch" conclusion for learning representations in DG (Ruan et al., 2021).

While attempting to restrict the set of feasible solutions, a DG algorithm, with its extra constraints, may as well reduce the grey area, by restricting the global optimal set to only solutions that also meet the constraints. With arbitrary constraints, there is a possibility that the grey area shrinks to null. Interestingly, a key insight from Theorem 3.8 is that, under the *Condition* 3.3, as long as the solution of an algorithm fulfills the *sufficient representation function* constraint (*Condition* 3.7), there exists a non-empty $\mathcal{F}^* \subseteq \mathcal{F}_{\mathcal{E}_{tr}}$; otherwise $\mathcal{F}^* = \emptyset$. In fact, that an algorithm meets a sufficient condition implies the satisfaction of *Condition* 3.7 by default.

**In summary**, an algorithm should be effectively designed to minimize the space $\mathcal{F}_{\mathcal{E}_{tr}}$ while maintaining the coverage of $\mathcal{F}^*$. *Condition* 3.3 ensures the *green* area is non-empty i.e., $\mathcal{F}_{\mathcal{E}_{tr}} \neq \emptyset$ while *Condition* 3.7 ensures the *grey* one is non-empty i.e., $\mathcal{F}^* \neq \emptyset$. Satisfying both conditions further guarantees the existence of the global optimal solutions in $\mathcal{F}^* \subseteq \mathcal{F}_{\mathcal{E}_{tr}}$. In contrast, if both conditions are violated, the algorithm has zero chance of achieving generalization. Despite its significance, existing DG algorithms tends to overlook *Condition* 3.7. From finite training domains, they thus cannot guarantee the possibility of searching for global optimal hypotheses.

## 4.2 UNDERSTANDING DG LITERATURE VIA NECESSITY

It is clear that in order to be considered a globally optimal candidate, a learned hypothesis from finite domains must meet the necessary conditions 3.3 and 3.7. Following the previous analysis, we here review the popular classes of DG methods and discuss when they meet or fail these conditions.

**Representation Alignment.** These approaches aim to learn a representation function $g$ for data $X$ such that $g(X)$ is invariant or consistent across different domains. Key studies like (Long et al., 2017; Ganin et al., 2016; Li et al., 2018b; Nguyen et al., 2021; Shen et al., 2018; Xie et al., 2017; Ilse et al., 2020) focus on learning such domain-invariant representations by reducing the divergence between latent marginal distributions $\mathbb{E}[g(X)|E]$ where $E$ represents a domain environment. Other methods seek to align the conditional distributions $\mathbb{E}[g(X)|Y = y, E]$ across domains as seen in (Li et al., 2018c; Tachet des Combes et al., 2020). However, achieving true invariance is challenging and can be excessively limiting. In some instances, improved alignment of features leads to greater joint errors (Johansson et al., 2019; Zhao et al., 2019; Phung et al., 2021).

**Theorem 4.2.** *(Johansson et al., 2019; Zhao et al., 2019; Phung et al., 2021) Distance between two marginal distribution $\mathbb{P}^e_{\mathcal{Y}}$ and $\mathbb{P}^{e'}_{\mathcal{Y}}$ can be upper-bounded:*

$$D\left(\mathbb{P}^e_{\mathcal{Y}}, \mathbb{P}^{e'}_{\mathcal{Y}}\right) \leq D\left(g_{\#}\mathbb{P}^e, g_{\#}\mathbb{P}^{e'}\right) + \mathcal{L}\left(f, \mathbb{P}^e\right) + \mathcal{L}\left(f, \mathbb{P}^{e'}\right)$$

*where $g_{\#}\mathbb{P}(X)$ denotes representation distribution on representation space $\mathcal{Z}$ induce by applying encoder with $g : \mathcal{X} \mapsto \mathcal{Z}$ on data distribution $\mathbb{P}$, $D$ can be $\mathcal{H}$-divergence (Zhao et al., 2019), Hellinger distance (Phung et al., 2021) or Wasserstein distance (Le et al., 2021) (Appendix A.2).*

Theorem 4.2 suggests that a substantial discrepancy in the label marginal distribution $D\left(\mathbb{P}^e_{\mathcal{Y}}, \mathbb{P}^{e'}_{\mathcal{Y}}\right)$ across training domains may result in strong *representation alignment* $D\left(g_{\#}\mathbb{P}^e, g_{\#}\mathbb{P}^{e'}\right)$ while increasing *domain-losses* $\left(\mathcal{L}\left(f, \mathbb{P}^e\right) + \mathcal{L}\left(f, \mathbb{P}^{e'}\right)\right)$. It's important to recognize that while the *representation alignment* strategy could challenge *Condition* 3.3, this alignment constraint can help reduce the cardinality of $\bigcap_{e \in \mathcal{E}_{tr}} \mathcal{F}_{\mathbb{P}^e}$. Thus, performance improvement is still attainable with careful adjustment of the alignment weight by exploiting the oracle knowledge of the target domain.

**Invariant Prediction.** These methods aim to learn a consistent optimal classifier across domains. For example, Invariant Risk Minimization (IRM) (Arjovsky et al., 2020) seeks to learn a representation function $g(x)$ with invariant predictors $\mathbb{E}[Y|g(x), E]$. This goal aligns with *Condition* 3.7 and encourages using invariant representations, without imposing restrictions that could affect *Condition* 3.3. VREx (Krueger et al., 2021) relaxes the IRM's constraint to enforce equal risks across domains, assuming that the optimal risks are similar across domains. If, however, the optimal solutions exhibit large loss variations, balancing risks could result in suboptimal performance for some domains, violating *Condition* 3.3. Furthermore, with a limited number of training domains, both IRM and VREx may struggle to identify the optimal invariant predictor, as discussed by Rosenfeld et al. (2020) and may not offer advantages over ERM, especially when representations from different domains occupy distinct regions in the representation space, as noted by (Ahuja et al., 2020).

IIB (Li et al., 2022a) and IB-IRM (Ahuja et al., 2021) integrate the information bottleneck principle with invariant prediction strategies. However, similar to IRM, these approaches only show benefits with a sufficient and diverse number of training domains. Otherwise, the information bottleneck even makes it susceptible to violating *Condition* 3.7. See Appendix B for further discussion.

**Augmentation.** Data augmentation (Mitrovic et al., 2020; Wang et al., 2022b; Shankar et al., 2018; Zhou et al., 2020; 2021; Xu et al., 2021; Zhang et al., 2017; Wang et al., 2020b; Zhao et al., 2020; Yao et al., 2022a; Carlucci et al., 2019; Yao et al., 2022b) have long been applied to DG. This strategy is to utilize predefined or learnable transformations $T$ on the original sample $X$ or its features $g(x)$ to create augmented data $T(X)$ or $T(g(x))$. Applying various transformations during training effectively increases the training dataset, which, according to Theorem 3.4, should narrow the hypothesis space. However, it's crucial that transformation $T$ maintains the integrity of the causal factors. This implies a *necessity for some knowledge of the target domain* to ensure the transformations do not alter the causal/invariant information (Gao et al., 2023), otherwise it risks violating *Condition* 3.7 (e.g., augmentation possibly introduces misleading information (Zhang & Ma, 2022)).

**Ensemble Learning.** Ensemble learning (Zhou, 2012) refers to training multiple copies of the same architecture with different initializations or splits of the training data, then ensembling the individual models for prediction. This straightforward technique has been shown to outperform a single model across various applications, including DG (Zhou et al., 2021; Ding & Fu, 2017; Zhou et al., 2021; Wang et al., 2020a; Mancini et al., 2018; Cha et al., 2021; Arpit et al., 2022). Unlike explicit ensemble methods where multiple models (or model components) need to be trained, Cha et al. (2021); Rame et al. (2022); Wortsman et al. (2022) demonstrate that averaging model weights (WA) at different time steps during training to form a single model at test time (Izmailov et al., 2018) can significantly enhance robustness under domain shift. Different from the previous works, our analysis in Section 5.1) provides a new insight that ensemble-based methods can also encourage the learning of *sufficient representation* (*Condition* 3.7) to promote generalizability.

## 5 SUFFICIENT INVARIANT REPRESENTATION LEARNING

Section 4.2 highlights that existing DG strategies attempt to maximize the likelihood of seeking a global optimal hypothesis from different directions yet with several drawbacks. Furthermore, that they all overlook *Condition* 3.7 poses a risk of landing in regions with empty solution set. Generally, an effective DG algorithm is one that strives to attain the sufficient conditions while guarantees the necessary conditions. Here we propose a method that exploits the joint effect of the two sets of conditions to boost generalization.

In the following, we explain how to incorporate the sufficient representation constraint via *ensemble learning* and present a novel *representation alignment* strategy that can enforce the necessary conditions. We particularly do not consider *invariant prediction* since it cannot substantiate its superiority over ERM with a potential of violating both necessary conditions. Meanwhile, *data augmentation* typically provides significant benefits and can be integrated in a plug-and-play fashion. Since it requires prior knowledge, users should apply it carefully based on their expertise.

### 5.1 SUFFICIENT REPRESENTATION CONSTRAINTS

By definition, a representation function $g$ is considered as sufficient representation if there exists a function $\phi \in \Phi$ such that: $\phi \circ g \in \mathcal{G}_c$. Our task can thus be translated into learning the representation $Z = g(X)$ that captures the most information about the causal factor $Z_c$. This motivates us to find $Z$ that maximizes the mutual information $I(Z; Z_c)$.

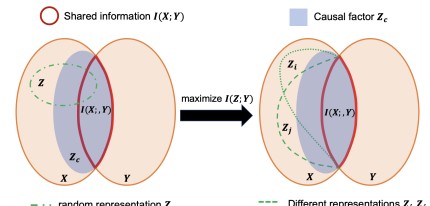

Given a specific domain, recall our model $Z \leftarrow X \leftarrow Z_c \rightarrow Y$, where $Y$ is influenced by $Z_c$ (the latent cause) and $Z_c$ also affects $X$. Note that $X$ is also under the influence of $Z_e$, which we omit here for simplicity. Since $Z_c$ is unob-

Figure 3: Information diagrams of $X, Y, Z_c$ and $Z = g(X)$. Learning multiple representations $Z_i, ..., Z_j$ through ensemble learning where $Z_i = g_i(X)$ s.t $g_i \in \{\arg\max_{g_i} I(g_i(X); Y)\}$ to maximize the shared information with $Z_c$.

served, we cannot directly measure or learn from it. However, we can leverage $Y$, which inherits the causal information of $Z_c$. This intuition can be best understood via an information diagram.

Let us examine Figure 3.(Left) that illustrates the mutual information of the 4 variables. We have $I(X, Y \mid Z_c) = 0$, meaning the causal features $Z_c$ must capture the shared information $I(X; Y)$. By Assumption 3.2 and Proposition 3.5, it follows that $X$ contains all information about $Z_c$.

By the chain rule of mutual information, we have that $I(Z; Z_c) \geq I(Z; Y)$. Thus, we resort to maximizing the lower bound $I(Z; Y)$ to increasing the chance of learning $Z$ that contains causal information $Z_c$. Recall that we use the cross-entropy loss $\ell : \mathcal{Y}_\Delta \times \mathcal{Y} \mapsto \mathbb{R}$ to optimize the hypothesis for training domains. It is well-known that minimizing the cross-entropy loss is equivalent to maximizing the lower bound of $I(Z; Y)$ (Qin et al., 2019; Colombo et al., 2021). In other words, hypotheses that are optimal on training domains (*Condition* 3.3) also promote the sufficient representation function condition (*Condition* 3.7). However, maximizing the lower bound $I(Z; Y)$ only ensures that $Z$ captures the shared information $I(X; Y)$ and potentially some additional information about $Z_c$ (as illustrated in Figure 3 (Right)).

To encourage the representation $Z$ to capture more information from $Z_c$, this approach can be extended to learn multiple versions of representations through ensemble learning. Specifically, we can learn an $M$-ensemble of representations $Z^M$:

$$Z^M = \left\{ Z_i = g_i(X) \mid g_i \in \arg\max_{g_i} I(g_i(X); Y) \right\}_{i=1}^M,$$

to capture as much information as possible about $Z_c$. This intuition aligns with the analysis of ensembles for OOD generalization presented in Rame et al. (2022).

## 5.2 SUBSPACE REPRESENTATION ALIGNMENT

*Representation Alignment* strategy helps reduce the cardinality of $\mathcal{F}_\cap$ but may compromise *Condition* 3.3 due to the potential trade-off between alignment constraints and domain losses (Theorem 4.2). **However, we now show that with a more careful design, we can address the trade-off effectively**. Our proposed strategy, called *Subspace Representation Alignment* (SRA), involves organizing training domains into distinct subspaces and aligning representations within these subspaces. This aims to diminish or completely remove differences in the marginal label distributions across these domains so that the search space can be reduced.

We consider *subspace projector* $\Gamma : \mathcal{X} \to \mathcal{M}$, given a subspace index $m \in \mathcal{M}$, we denote $A_m = \Gamma^{-1}(m) = \{x : \Gamma(x) = m\}$ is the region on data space which has the same index $m$. Let $\mathbb{P}_m^e$ be the distribution restricted by $\mathbb{P}^e$ over the set $A_m$. Eventually, we define $\mathbb{P}_m^e (y \mid x)$ as the probabilistic labeling distribution on the subspace $(A_m, \mathbb{P}_m^e)$, meaning that if $x \sim \mathbb{P}_m^e$, $\mathbb{P}_m^e (y \mid x) = \mathbb{P}_e (y \mid x)$. Since each data point $x \in \mathcal{X}$ corresponds to only a single $\Gamma(x)$, the data space is partitioned into disjoint sets, i.e., $\mathcal{X} = \bigcup_{m=1}^{\mathcal{M}} A_m$, where $A_m \cap A_n = \emptyset, \forall m \neq n$. Consequently, $\mathbb{P}^e := \sum_{m \in \mathcal{M}} \pi_m^e \mathbb{P}_m^e$ where $\pi_m^e = \mathbb{P}^e (A_m) / \sum_{m' \in \mathcal{M}} \mathbb{P}^e (A_{m'})$.

**Theorem 5.1.** *Given a subspace projector $\Gamma$, if the loss function $\ell$ is upper-bounded by a positive constant $L$, then: (i) The target general loss is upper-bounded:*

$$|\mathcal{E}_{tr}| \sum_{e \in \mathcal{E}_{tr}} \mathcal{L}(f, \mathbb{P}^e) \leq \sum_{e \in \mathcal{E}_{tr}} \sum_{m \in \mathcal{M}} \pi_m^e \mathcal{L}(f, \mathbb{P}_m^e) + L \sum_{e,e' \in \mathcal{E}_{tr}} \sum_{m \in \mathcal{M}} \pi_m^e D\left(g_\# \mathbb{P}_m^e, g_\# \mathbb{P}_m^{e'}\right),$$

*(ii) Distance between two label marginal distribution $\mathbb{P}_m^e(Y)$ and $\mathbb{P}_m^{e'}(Y)$ can be upper-bounded:*

$$D\left(\mathbb{P}_{\mathcal{Y},m}^e, \mathbb{P}_{\mathcal{Y},m}^{e'}\right) \leq D\left(g_\# \mathbb{P}_m^e, g_\# \mathbb{P}_m^{e'}\right) + \mathcal{L}(f, \mathbb{P}_m^e) + \mathcal{L}\left(f, \mathbb{P}_m^{e'}\right)$$

*where $g_\# \mathbb{P}$ denotes representation distribution on $\mathcal{Z}$ induce by applying $g$ with $g : \mathcal{X} \mapsto \mathcal{Z}$ on data distribution $\mathbb{P}$, $D$ can be $\mathcal{H}$-divergence, Hellinger or Wasserstein distance. (Proof in Appendix A.12)*

In Theorem 5.1, (i) illustrates that *domain-specific losses* can be broken down into *losses* and *representation alignments* within individual subspaces. Optimizing the subspace-specific losses across domains ensures optimizing the overall loss within the original domains are optimized. Meanwhile, (ii) demonstrates that the distance between the marginal label distributions is now grounded within

subspaces, denoted as $d_{1/2}\left(\mathbb{P}^e_{\mathcal{Y},m}, \mathbb{P}^{e'}_{\mathcal{Y},m}\right)$. Theorem 5.1 suggests that appropriately distributing training domains across subspaces can reduce both the upper and lower bounds. Particularly, for a given subspace index $m$, if $D\left(\mathbb{P}^e_{\mathcal{Y},m}, \mathbb{P}^{e'}_{\mathcal{Y},m}\right) = 0$, we can jointly optimize both *domains losses* $\mathcal{L}\left(f, \mathbb{P}^e_m\right) + \mathcal{L}\left(f, \mathbb{P}^{e'}_m\right)$ and *representation alignment* $D\left(g_\# \mathbb{P}^e_m, g_\# \mathbb{P}^{e'}_m\right)$. Consequently, optimizing the RHS of (ii) for all supspaces is equivalent to minimizing the RHS of (i).

The question now is how we can manage the training distribution into a subspace such that $D\left(\mathbb{P}^e_{\mathcal{Y},m}, \mathbb{P}^{e'}_{\mathcal{Y},m}\right)$ is reduced, potentially even to zero. Fortunately, working within training domains, we anticipate that $f \in \cap_{e \in \mathcal{E}_{tr}} \mathcal{F}_{\mathbb{P}^e}$ will predict the ground truth label $f(x) = f^*(x)$ where $f^* \in \mathcal{F}^*$. We can define a projector $\Gamma = f$, which induces a set of subspace indices $\mathcal{M} = \{m = \hat{y} \mid \hat{y} = f(x), x \in \bigcup_{e \in \mathcal{E}_{tr}} \text{supp} \mathbb{P}^e\} \subseteq \Delta_{|\mathcal{Y}|}$. As a result, given subspace index $m \in \mathcal{M}, \forall i \in \mathcal{Y}, \mathbb{P}^e_{\mathcal{Y},m}(Y = i) = \mathbb{P}^{e'}_{\mathcal{Y},m}(Y = i) = \sum_{x \in f^{-1}(m)} \mathbb{P}(Y = i \mid x) = m[i]$. Consequently, $D\left(\mathbb{P}^e_{\mathcal{Y},m}, \mathbb{P}^{e'}_{\mathcal{Y},m}\right) = 0$ for all $m \in \mathcal{M}$, allowing us to jointly optimize both *domain losses* and *representation alignment*.

The **final optimization objective**, encapsulating the constraints of optimal hypothesis for all training domain, ensemble for sufficient representation, and subspace representation alignment is given by:

$$\min_f \underbrace{\sum_{e,e' \in \mathcal{E}_{tr}} \sum_{m \in \mathcal{M}} D\left(g\#\mathbb{P}^e_m, g\#\mathbb{P}^{e'}_m\right)}_{\text{Subspace Representation Alignment}} \text{ s.t. } f \in \underbrace{\bigcap_{e \in \mathcal{E}_{tr}} \underset{f}{\arg\min} \mathcal{L}\left(f, \mathbb{P}^e\right)}_{\text{Training domain optimal hypothesis}} \quad (4)$$

where $\mathcal{M} = \{\hat{y} \mid \hat{y} = f(x), x \in \bigcup_{e \in \mathcal{E}_{tr}} \text{supp} \mathbb{P}^e\}$ and $D$ can be $\mathcal{H}$-divergence, Hellinger distance, Wasserstein distance. We provide the details on the practical implementation of the proposed objective in Appendix C.

## 5.3 EXPERIMENTS

In this section, we present empirical evidence validating our theoretical takeaways, that is enforcing good sufficient conditions (SRA) while encouraging necessary conditions (Ensemble) can improve generalization. For the ensemble component, we utilize the weight averaging strategy from the SWAD method (Cha et al., 2021) for efficient inference. Importantly, our analysis highlights that using ensembles for targeting the sufficient representation constraint can provide crucial benefits for generalization. This strategy should therefore not be viewed as merely post-processing or an orthogonal technique in DG setting.

Table 2 compares our method against two popular representation alignment strategies: DANN and CDANN, on 5 datasets from DomainBed benchmark Gulrajani & Lopez-Paz (2021). Note that we also use $\mathcal{H}$-divergence for alignment in DANN and CDANN. The only difference is that DANN aligns the whole domain representation, CDANN aligns class-conditional representation, while SRA employs subspace-conditional alignment. First, it is seen that both DANN and CDANN cannot surpass ERM overall with and without SWAD. This supports our analysis in Section 4.2 that these methods violate the necessary condition. In contrast, our method consistently achieves better performance than the baseline approaches on all datasets. We further demonstrate the benefit of an ensemble approach by averaging the predictions of models trained with different random seeds (SRA + SWAD + Ensemble), resulting in a performance boost. Full experimental results and detailed settings are provided in Appendix D.

## 6 LIMITATIONS AND CONCLUSION

This paper presents a comprehensive study of existing DG algorithms under various conditions towards achieving global optimal hypothesis. While the condition of *sufficient representation* is often overlooked in DG literature, its role is critical to understanding whether a DG algorithm truly generalizes, underscoring several facets of generalization that current benchmarking fails to factor in. Providing a theoretical guarantee for the verifiability of many of the conditions under analysis is

Table 2: Classification accuracy (%) for all algorithms across datasets.

| Algorithm | VLCS | PACS | OfficeHome | TerraIncognita | DomainNet | Avg |
|---|---|---|---|---|---|---|
| ERM (Gulrajani & Lopez-Paz, 2021) | $77.5 \pm 0.4$ | $85.5 \pm 0.2$ | $66.5 \pm 0.3$ | $46.1 \pm 1.8$ | $40.9 \pm 0.1$ | 63.3 |
| DANN (Ganin et al., 2016) | $78.6 \pm 0.4$ | $83.6 \pm 0.4$ | $65.9 \pm 0.6$ | $46.7 \pm 0.5$ | $38.3 \pm 0.1$ | 62.6 |
| CDANN (Li et al., 2018b) | $77.5 \pm 0.1$ | $82.6 \pm 0.9$ | $65.8 \pm 1.3$ | $45.8 \pm 1.6$ | $38.3 \pm 0.3$ | 62.0 |
| **Ours** (SRA) | $76.4 \pm 0.7$ | $86.3 \pm 1.1$ | $66.4 \pm 0.7$ | $49.5 \pm 1.0$ | $44.5 \pm 0.3$ | 64.6 |
| SWAD (Cha et al., 2021) | $79.1 \pm 0.4$ | $88.1 \pm 0.4$ | $70.6 \pm 0.3$ | $50.0 \pm 0.4$ | $46.5 \pm 0.2$ | 66.9 |
| SWAD + DANN | $79.2 \pm 0.0$ | $87.9 \pm 0.5$ | $70.5 \pm 0.1$ | $50.6 \pm 0.6$ | $45.7 \pm 0.1$ | 66.8 |
| SWAD + CDANN | $79.3 \pm 0.2$ | $87.7 \pm 0.3$ | $70.4 \pm 0.1$ | $50.7 \pm 0.1$ | $45.7 \pm 0.2$ | 66.8 |
| **Ours** (SRA + SWAD) | $\underline{79.4} \pm 0.4$ | $\underline{88.7} \pm 0.2$ | $\underline{72.1} \pm 0.5$ | $\underline{51.6} \pm 1.2$ | $\underline{47.6} \pm 0.1$ | $\underline{67.9}$ |
| **Ours** (SRA + SWAD + Ensemble) | $\mathbf{79.8} \pm 0.0$ | $\mathbf{89.2} \pm 0.0$ | $\mathbf{73.2} \pm 0.0$ | $\mathbf{52.2} \pm 0.0$ | $\mathbf{48.7} \pm 0.6$ | **68.6** |

beyond the scope of the current work. We here at best draw insights from our analysis to translate the conditions into practical constraints for optimization. Our future works will also focus on designing an evaluation framework that can characterize necessary and sufficient conditions as well as quantify the likelihood of achieving generalization.

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

# A  THEORETICAL DEVELOPMENT

In this section, we present all the proofs of our theoretical development.

## A.1  NECESSARY AND SUFFICIENT CONDITIONS FOR ACHIEVING GENERALIZATION

For readers' convenience, we recapitulate our definition and assumptions:

*Domain objective*: Given a domain $\mathbb{P}^e$, let the hypothesis $f : \mathcal{X} \to \Delta_{|\mathcal{Y}|}$ is a map from the data space $\mathcal{X}$ to the the $C$-simplex label space $\Delta_{|\mathcal{Y}|} := \left\{ \alpha \in \mathbb{R}^{|\mathcal{Y}|} : \|\alpha\|_1 = 1 \wedge \alpha \geq 0 \right\}$. Let $l : \mathcal{Y}_\Delta \times \mathcal{Y} \mapsto \mathbb{R}$ be a loss function, where $\ell\left(f\left(x\right), y\right)$ with $f\left(x\right) \in \mathcal{Y}_\Delta$ and $y \in \mathcal{Y}$ specifies the loss (i.e., cross-entropy) to assign a data sample $x$ to the class $y$ by the hypothesis $f$. The general loss of the hypothesis $f$ w.r.t. a given domain $\mathbb{P}^e$ is:

$$\mathcal{L}\left(f, \mathbb{P}^e\right) := \mathbb{E}_{(x,y) \sim \mathbb{P}^e}\left[\ell\left(f\left(x\right), y\right)\right]. \tag{5}$$

**Assumption A.1.** (Label-identifiability). We assume that for any pair $z_c, z_c' \in \mathcal{Z}_c$, $\mathbb{P}(Y|Z_c = z_c) = \mathbb{P}(Y|Z_c = z_c')$ if $\psi_x(z_c, z_e, u_x) = \psi_x(z_c', z_e', u_x')$ for some $z_e, z_e', u_x, u_x'$.

**Assumption A.2.** (Causal support). We assume that the union of the support of causal factors across training domains covers the entire causal factor space $\mathcal{Z}_c$: $\cup_{e \in \mathcal{E}_{tr}} \mathrm{supp}\{\mathbb{P}^e\left(Z_c\right)\} = \mathcal{Z}_c$ where $\mathrm{supp}(\cdot)$ specifies the support set of a distribution.

**Corollary A.3.** $\mathcal{F} \neq \emptyset$ *if and only if Assumption A.1 holds.*

*Proof.* The **"if"** direction is directly derived from the Proposition A.4. We prove **"only if"** direction by contraction.

If Assumption A.1 does not hold, there a pair $x = x'$ such that $x = \psi_x(z_c, z_e, u_x)$ $x' = \psi_x(z_c', z_e', u_x')$ for some $z_e, z_e', u_x, u_x'$ and $\mathbb{P}(Y|Z_c = z_c) \neq \mathbb{P}(Y|Z_c = z_c')$.

By definition of $f \in \mathcal{F}^*$, $f(x) = \mathbb{P}(Y|Z_c = z_c) \neq \mathbb{P}(Y|Z_c = z_c') = f(x') = f(x)$ which is a contradiction. (It is worth noting that a domain containing only one sample $x$ is also valid within our data-generation process depicted in Figure 1.). $\square$

**Proposition A.4.** *(Invariant Representation Function) Under Assumption.A.1, there exists a set of deterministic representation function $(\mathcal{G}_c \neq \emptyset) \in \mathcal{G}$ such that for any $g \in \mathcal{G}_c$, $\mathbb{P}(Y \mid g(x)) = \mathbb{P}(Y \mid z_c)$ and $g(x) = g(x')$ holds true for all $\{(x, x', z_c) \mid x = \psi_x(z_c, z_e, u_x), x' = \psi_x(z_c, z_e', u_x') \text{ for all } z_e, z_e', u_x, u_x'\}$*

*Proof.* Under Assumption.A.1, we can always choose a deterministic function $g_c : \mathcal{X} \to \mathcal{Z}_c$ such that the outcome of $g_c(x)$, can be any $z_c \in \{z_c \mid x = \psi_x(z_c, z_e, u_x)\}$ and $\mathbb{P}(Y \mid g_c(x)) = \mathbb{P}(Y \mid z_c)$, will consistently provide an accurate prediction of $Y$. In essence, Y is identifiable over the pushforward measure $g_c \# \mathbb{P}(X)$.

$\square$

**Corollary A.5.** *(Invariant Representation Function Properties) For any $g \in \mathcal{G}_c$, the following properties hold:*

1. *$g$ is a mapping function directly from the sample space $\mathcal{X}$ to the causal feature space $\mathcal{Z}_c$, such that $g : \mathcal{X} \to \mathcal{Z}_c$.*

2. *Given a deterministic equivalent causal transformation mapping $T : \mathcal{Z}_c \to \mathcal{Z}_c$, which maps a causal factor $z_c$ to another equivalent causal factor $T(z_c)$, such that $\mathbb{P}(Y \mid z_c) = \mathbb{P}(Y \mid T(z_c))$, then we have $g(x) = T(z_c)$ holds for all $\{x \mid x = \psi_x(z_c, z_e, u_x), \text{ for all } z_e, u_x\}$.*

3. *Given $\ell$ is the Cross-Entropy Loss i.e., $\ell(h(z_c), y) = -\sum_{y \in \mathcal{Y}} \mathbb{P}(Y = y \mid z_c) \log h(z_c)[y]$, there exists $h^*$ such that:*

$$h^* \in \bigcap_{z_c \in \mathcal{Z}_c} \underset{h \in \mathcal{H}}{argmin} \, \mathbb{E}_{y \sim \mathbb{P}(Y|z_c)} \ell\left(h(z_c), y\right),$$

*Proof.* We prove each property as follows:

*Proof of property-1:* Suppose there exists $g : \mathcal{X} \to \mathcal{Z}$ such that $\mathbb{P}(Y \mid g(x)) = \mathbb{P}(Y \mid z_c)$ holds true for all $\{(x, z_c) \mid x = \psi_x(z_c, z_e, u_x)$ for all $z_e, u_x\}$.

If $g$ is not a function from $\mathcal{X}$ to $\mathcal{Z}_c$, then $g(x)$ may include spurious features $z_e$, or both $z_c$ and $z_e$ for $x = \psi(z_c, z_e, u_x)$.

Based on the structural causal model (SCM) depicted in Figure 1, it follows that $Z_e \not\perp Y$, meaning that the environmental feature $Z_e$ is spuriously correlated with $Y$. Consequently,

$$\mathbb{P}(Y \mid g(x = \psi(z_c, z_e, u_x))) \neq \mathbb{P}(Y \mid g(x = \psi(z_c, z_e', u_x)))$$

for some $z_e \neq z_e'$, which is a contradiction.

*Proof of property-2:* Since $g : \mathcal{X} \to \mathcal{Z}_c$ and $\mathbb{P}(Y \mid g(x)) = \mathbb{P}(Y \mid z_c)$ holds true for all $\{(x, z_c) \mid x = \psi_x(z_c, z_e, u_x)$ for all $z_e, u_x\}$, the outcome of $g(x)$ have to be any $z_c' \in \mathcal{Z}_c$ such that $\mathbb{P}(Y \mid z_c) = \mathbb{P}(Y \mid z')$, which means $g(x) = T(z_c)$ holds for $\{x \mid x = \psi_x(z_c, z_e, u_x)\}$

This highlights the flexibility of the family of invariant representation functions $\mathcal{G}_c$, as they allow the model to map a sample $x = \psi(z_c, z_e, u_x)$ to a set of equivalent causal factors $\{z_c' \in \mathcal{Z}_c \mid \mathbb{P}(Y \mid z_c) = \mathbb{P}(Y \mid z_c')\}$, rather than requiring an exact mapping to $z_c$.

Finally, since $g(x) = g(x')$ holds true for all $\{(x, x', z_c) \mid x = \psi_x(z_c, z_e, u_x), x' = \psi_x(z_c, z_e', u_x')$ for all $z_e, z_e', u_x, u_x'\}$, $g(x) = T(z_c)$ holds for all $\{x \mid x = \psi_x(z_c, z_e, u_x)$, for all $z_e, u_x\}$

*Proof of property-3:*  $\qquad\qquad\qquad\qquad\qquad\qquad\qquad\qquad\qquad\qquad\qquad\qquad$ $\square$

Given $z_c \in \mathcal{Z}_c$ and $\ell(h(z_c), y) = -\sum_{y \in \mathcal{Y}} \mathbb{P}(Y = y \mid z_c) \log h(z_c)[y]$, it is easy to show that the optimal

$$h^* = \operatorname*{argmin}_{h \in \mathcal{H}} \mathbb{E}_{y \sim \mathbb{P}(Y \mid z_c)} \ell\left(h(z_c), y\right)$$

is the conditional probability distribution $h^*(z_c) = \mathbb{P}(Y \mid z_c)$.

Based on structural causal model (SCM) depicted in Figure 1, $\mathbb{P}(Y \mid z_c)$ remains stable across all domains. Therefore, there exists an optimal function $h^*$ such that:

$$h^* \in \bigcap_{z_c \in \mathcal{Z}_c} \operatorname*{argmin}_{h \in \mathcal{H}} \mathbb{E}_{y \sim \mathbb{P}(Y \mid z_c)} \ell\left(h(z_c), y\right),$$

where $h^*(z_c) = \mathbb{P}(Y \mid z_c)$ for all $z_c \in \mathcal{Z}_c$

**Theorem A.6.** *(Theorem 3.6 in the main paper) Denote the set of **domain optimal hypotheses** of $\mathbb{P}^e$ induced by $g \in \mathcal{G}$:*

$$\mathcal{F}_{\mathbb{P}^e, g} = \left\{ h \circ g \mid h \in \operatorname*{argmin}_{h' \in \mathcal{H}} \mathcal{L}\left(h' \circ g, \mathbb{P}^e\right) \right\}.$$

*If $\mathrm{supp}\{\mathbb{P}^e(Z_c)\} = \mathcal{Z}_c$ and $g \in \mathcal{G}_c$, then $\mathcal{F}_{\mathbb{P}^e, g} \subseteq \mathcal{F}^*$.*

*Proof.* Given $\mathrm{supp}\{\mathbb{P}^e(Z_c)\} = \mathcal{Z}_c$ and $g_c \in \mathcal{G}_c$, it suffices to prove that for any $f_c = h_c \circ g_c \in \mathcal{F}_{\mathbb{P}^e, g_c}$, we have:

$$f_c \in \bigcap_{\mathbb{P}^e \in \mathcal{P}} \operatorname*{argmin}_{f \in \mathcal{F}} \mathcal{L}\left(f, \mathbb{P}^e\right). \qquad (6)$$

To prove (6), we only need to show that for any $f = h \circ g_c \in \mathcal{F}$ and $\mathbb{P}^{e'} \in \mathcal{P}$:

$$\mathcal{L}\left(f, \mathbb{P}^{e'}\right) \geq \mathcal{L}\left(f_c, \mathbb{P}^{e'}\right), \qquad (7)$$

which is equivalent to:

$$\mathbb{E}_{(x,y)\sim\mathbb{P}^{e'}}\left[\ell\left(f\left(x\right),y\right)\right]\geq\mathbb{E}_{(x,y)\sim\mathbb{P}^{e'}}\left[\ell\left(f_{c}\left(x\right),y\right)\right]. \tag{8}$$

*Step 1: Simplifying the general loss using the invariant representation function $g_c$.*

Based on structural causal model (SCM) depicted in Figure 1 we have a distribution (domain) over the observed variables $(X,Y)$ given the environment $E=e\in\mathcal{E}$:

$$\mathbb{P}^e(X,Y) = \int_{\mathcal{Z}_c}\int_{\mathcal{Z}_e}\mathbb{P}^e(X,Y,Z_c=z_c,Z_e=z_e)d_{z_c}d_{z_e}$$

$$= \int_{\mathcal{Z}_c}\int_{\mathcal{Z}_e}\mathbb{P}^e(X,Y,z_c,z_e)d_{z_c}d_{z_e}$$

$$= \int_{\mathcal{Z}_c}\int_{\mathcal{Z}_e}\mathbb{P}^e(X\mid z_c,z_e)\mathbb{P}^e(Y\mid z_c)\mathbb{P}^e(z_c)\mathbb{P}^e(z_e)d_{z_c}d_{z_e}$$

$$= \int_{\mathcal{Z}_c}\int_{\mathcal{Z}_e}\mathbb{P}^e(z_c)\mathbb{P}^e(z_e)\int_{\mathcal{X}}\mathbb{P}^e(X=x\mid z_c,z_e)\mathbb{P}^e(Y\mid z_c)d_{z_c}d_{z_e}d_x$$

$$= \int_{\mathcal{Z}_c}\int_{\mathcal{Z}_e}\mathbb{P}^e(z_c)\mathbb{P}^e(z_e)\int_{\mathcal{X}}\mathbb{P}^e(X=x\mid z_c,z_e)\int_{\mathcal{Y}}\mathbb{P}^e(Y=y\mid z_c)d_{z_c}d_{z_e}d_xd_y$$

$$= \int_{\mathcal{Z}_c}\int_{\mathcal{Z}_e}\mathbb{P}^e(z_c)\mathbb{P}^e(z_e)\int_{\mathcal{X}}\int_{\mathcal{U}_x}\mathbb{P}^e(X=x\mid z_c,z_e,u_x)\mathbb{P}^e(u_x)\int_{\mathcal{Y}}\mathbb{P}^e(Y=y\mid z_c)d_{z_c}d_{z_e}d_xd_yd_{u_x}$$

$$\overset{(1)}{=} \int_{\mathcal{Z}_c}\int_{\mathcal{Z}_e}\mathbb{P}^e(z_c)\mathbb{P}^e(z_e)\int_{\mathcal{X}}\int_{\mathcal{U}_x}\mathbb{I}_{x=\psi_x(z_c,z_e,u_x)}\mathbb{P}^e(u_x)\int_{\mathcal{Y}}\mathbb{P}^e(Y=y\mid z_c)d_{z_c}d_{z_e}d_xd_yd_{u_x}$$

We have $\overset{(1)}{=}$ by definition of SCM, $x$ is the deterministic function of $(z_c,z_e,u_x)$.

Therefore we have:

$$\mathbb{E}_{(x,y)\sim\mathbb{P}^e(X,Y)}\left[\ell\left(f\left(x\right),y\right)\right]$$

$$= \int_{\mathcal{Z}_c}\int_{\mathcal{Z}_e}\mathbb{P}^e(z_c)\mathbb{P}^e(z_e)\int_{\mathcal{X}}\int_{\mathcal{U}_x}\mathbb{I}_{x=\psi_x(z_c,z_e,u_x)}\mathbb{P}^e(u_x)\int_{\mathcal{Y}}\mathbb{P}^e(Y=y\mid z_c)\ell\left(f\left(x\right),y\right)d_{z_c}d_{z_e}d_xd_yd_{u_x}$$

$$= \int_{\mathcal{Z}_c}\int_{\mathcal{Z}_e}\mathbb{P}^e(z_c)\mathbb{P}^e(z_e)\int_{\mathcal{U}_x}\int_{\mathcal{Y}}\mathbb{P}^e(Y=y\mid z_c)\int_{\mathcal{X}}\mathbb{I}_{x=\psi_x(z_c,z_e,u_x)}\ell\left(f\left(x\right),y\right)\mathbb{P}^e(u_x)d_{z_c}d_{z_e}d_xd_yd_{u_x}$$

$$= \int_{\mathcal{Z}_c}\int_{\mathcal{Z}_e}\mathbb{P}^e(z_c)\mathbb{P}^e(z_e)\int_{\mathcal{U}_x}\int_{\mathcal{Y}}\mathbb{P}^e(Y=y\mid z_c)\int_{\mathcal{X}}\mathbb{I}_{x=\psi_x(z_c,z_e,u_x)}\ell\left(f\left(\psi_x(z_c,z_e,u_x)\right),y\right)\mathbb{P}^e(u_x)d_{z_c}d_{z_e}d_xd_yd_{u_x}$$

$$= \int_{\mathcal{Z}_c}\int_{\mathcal{Z}_e}\mathbb{P}^e(z_c)\mathbb{P}^e(z_e)\int_{\mathcal{U}_x}\int_{\mathcal{Y}}\mathbb{P}^e(Y=y\mid z_c)\ell\left(f\left(\psi_x(z_c,z_e,u_x)\right),y\right)\mathbb{P}^e(u_x)d_{z_c}d_{z_e}d_yd_{u_x}$$

$$= \int_{\mathcal{Z}_c}\int_{\mathcal{Z}_e}\mathbb{P}^e(z_c)\mathbb{P}^e(z_e)\int_{\mathcal{U}_x}\mathbb{E}_{y\sim\mathbb{P}(Y\mid z_c)}\left[\ell\left(f\left(\psi_x(z_c,z_e,u_x)\right),y\right)\right]\mathbb{P}^e(u_x)d_{z_c}d_{z_e}d_{u_x}$$

$$= \int_{\mathcal{Z}_c}\int_{\mathcal{Z}_e}\mathbb{P}^e(z_c)\mathbb{P}^e(z_e)\int_{\mathcal{U}_x}\mathbb{E}_{y\sim\mathbb{P}(Y\mid z_c)}\left[\ell\left(\left(h\circ g_c\right)\left(\psi_x(z_c,z_e,u_x)\right),y\right)\right]\mathbb{P}^e(u_x)d_{z_c}d_{z_e}d_{u_x}$$

$$\overset{(1)}{=} \int_{\mathcal{Z}_c}\int_{\mathcal{Z}_e}\mathbb{P}^e(z_c)\mathbb{P}^e(z_e)\int_{\mathcal{U}_x}\mathbb{E}_{y\sim\mathbb{P}(Y\mid z_c)}\left[\ell\left(h\left(T(z_c)\right),y\right)\right]\mathbb{P}^e(u_x)d_{z_c}d_{z_e}d_{u_x}$$

$$= \int_{\mathcal{Z}_c}\mathbb{P}^e(z_c)\mathbb{E}_{y\sim\mathbb{P}(Y\mid z_c)}\left[\ell\left(h\left(T(z_c)\right),y\right)\right]d_{z_c}$$

$$= \int_{\mathcal{Z}_c}\mathbb{P}^e(z_c)\mathbb{E}_{y\sim\mathbb{P}(Y\mid T(z_c))}\left[\ell\left(h\left(T(z_c)\right),y\right)\right]d_{z_c}$$

$$\overset{(2)}{=} \int_{\mathcal{Z}_c}T_\#\mathbb{P}^e(z_c)\mathbb{E}_{y\sim\mathbb{P}(Y\mid z_c)}\left[\ell\left(h\left(z_c\right),y\right)\right]d_{z_c}$$

We have:

- $\overset{(1)}{=}$ by property-2 of $g_c$ (Corollary A.5);

- $\overset{(2)}{=}$ because $T : \mathcal{Z}_c \to \mathcal{Z}_c$ and $T_\# \mathbb{P}^e(z_c) = \int_{z_c' \in T^{-1}(z_c)} \mathbb{P}^e(z_c') dz_c'$

Now, to prove (8), we only need to show:

$$\int_{\mathcal{Z}_c} T_\# \mathbb{P}^{e'}(z_c) \mathbb{E}_{y \sim \mathbb{P}(Y|z_c)} \left[ \ell\left( h_c\left( z_c \right), y \right) \right] dz_c \leq \int_{\mathcal{Z}_c} T_\# \mathbb{P}^{e'}(z_c) \mathbb{E}_{y \sim \mathbb{P}(Y|z_c)} \left[ \ell\left( h\left( z_c \right), y \right) \right] dz_c \quad (9)$$

*Step 2: Generalization of $h_c$. Step-1* Demonstrate that $h_c$ only needs to make predictions for the set of causal factors $z_c \in \mathcal{Z}_c$. Therefore, it is sufficient to show that $h_c$ is optimal for every $z \in \mathcal{Z}_c$.

Recall that $f_c = h_c \circ g_c \in \mathcal{F}_{\mathbb{P}^e, g_c}$, therefore,

$$h_c \in \underset{h \in \mathcal{H}}{\operatorname{argmin}} \int_{\mathcal{Z}_c} T_\# \mathbb{P}^e(z_c) \mathbb{E}_{y \sim \mathbb{P}(Y|z_c)} \left[ \ell\left( h\left( z_c \right), y \right) \right] dz_c$$

By property-3 of $g_c$ (Corollary A.5), there exists an optimal function $h^*$ such that:

$$h^* \in \bigcap_{z_c \in \mathcal{Z}_c} \underset{h \in \mathcal{H}}{\operatorname{argmin}} \, \mathbb{E}_{y \sim \mathbb{P}(Y|z_c)} \ell\left( h(z_c), y \right),$$

Property-3 of $g_c$ ensures the existence of an optimal $h^*$ for every causal factor $z_c \in \mathcal{Z}_c$, it follows that $h_c$ must also be optimal for every causal feature $z_c$ within its support, $\operatorname{supp} \mathbb{P}^e(Z_e)$. This implies that $h_c(z_c) = h^*(z_c)$ for every $z_c$ where $\mathbb{P}^e(z_e) > 0$.

Moreover, since $\operatorname{supp} \mathbb{P}^e(Z_e) = \mathcal{Z}_c$, this implies that $h_c(z_c) = h^*(z_c)$ for every $z_c \in \mathcal{Z}_c$.

*Step-3: Proof of (9).*

$$\int_{\mathcal{Z}_c} T_\# \mathbb{P}^e(z_c) \mathbb{E}_{y \sim \mathbb{P}(Y|z_c)} \left[ \ell\left( h_c\left( z_c \right), y \right) \right] dz_c \leq \int_{\mathcal{Z}_c} T_\# \mathbb{P}^{e'}(z_c) \mathbb{E}_{y \sim \mathbb{P}(Y|z_c)} \left[ \ell\left( h\left( z_c \right), y \right) \right] dz_c$$

From *step-2*, we have

$$\mathbb{E}_{y \sim \mathbb{P}(Y|z_c)} \left[ \ell\left( h_c\left( z_c \right), y \right) \right] \leq \mathbb{E}_{y \sim \mathbb{P}(Y|z_c)} \left[ \ell\left( h\left( z_c \right), y \right) \right]$$

for all $z_c \in \mathcal{Z}_c$. By taking the expectation and applying the law of iterated expectation, inequality (9) follows. This concludes the proof.

$\square$

**Theorem A.7.** *(**Theorem 3.8 in the main paper**) Considering the training domains $\mathbb{P}^e$ and representation function $g$, let $\mathcal{H}_{\mathbb{P}^e, g} = \underset{h \in \mathcal{H}}{\operatorname{argmin}} \, \mathcal{L}\left( h \circ g, \mathbb{P}^e \right)$ represent the set of optimal classifiers on $g \# \mathbb{P}^e$ (the push-forward distribution by applying $g$ on $\mathbb{P}^e$), **the best generalization classifier** from $\mathbb{P}^e$ to $\mathcal{P}$ is defined as*

$$\mathcal{F}_{\mathbb{P}^e, g}^B = \left\{ h \circ g \mid h = \underset{h' \in \mathcal{H}_{\mathbb{P}^e, g}}{\operatorname{argmin}} \, \underset{e' \in \mathcal{E}}{\sup} \, \mathcal{L}\left( h' \circ g, \mathbb{P}^{e'} \right) \right\} \quad (10)$$

*Give representation function $g : \mathcal{X} \to \mathcal{Z}$ then $\forall \mathbb{P}^e \sim \mathcal{P}$ we have $\mathcal{F}_{\mathbb{P}^e, g}^B \subseteq \mathcal{F}^*$ if and only if $g \in \mathcal{G}_s$.*

*Proof.* We first proof **"if"** direction. If $g \in \mathcal{G}_s$, we have:

1. There exists a function $\phi$ such that $\phi \circ g \in \mathcal{G}_c$, which implies the existence of a $g_c \in \mathcal{G}_c$ such that $\phi \circ g = g_c$.

2. By the definition of $g_c \in \mathcal{G}_c$, we can always find a classifier $h$ such that $h \circ g_c \in \mathcal{F}^*$.

Recall that the definition of $\mathcal{F}^B_{\mathbb{P}^e, g}$ implies that, given $g$, we perform an oracle search for a classifier $h' \in \mathcal{H}_{\mathbb{P}^e, g}$ such that $h' \circ g$ achieves the best generalization across any domain $e' \in \mathcal{E}$.

From (1) and (2) we have $h \circ g_c = h \circ \phi \circ g \in \mathcal{F}^*$. Therefore, we can construct classifier $h_\phi = h \circ \phi$, then $h_\phi \circ g = h \circ \phi \circ g = h \circ g_c \in \mathcal{F}^*$.

Since $h_\phi \circ g \in \mathcal{F}^*$ is the optimal hypothesis across all domains, it is also the optimal hypothesis for the specific domain $\mathbb{P}^e$, i.e., $h_\phi \circ g \in \mathcal{F}_{\mathbb{P}^e, g}$. This implies that $h_\phi \in \mathcal{H}_{\mathbb{P}^e, g}$. Consequently, the set $\mathcal{F}^B_{\mathbb{P}^e, g} \subseteq \mathcal{F}^*$.

We will prove **"only if"** direction by contraction. We show that if $g$ is not sufficient-representation, there exists multiple target domains where learned classifier $h \in \mathcal{F}^*_g$ on $g$ performs arbitrarily bad.

If there does not exist a function $\phi$ such that $\phi \circ g \in \mathcal{G}_c$, then for any $\phi$ and $\forall (h, h_c)$ where $h \circ g \in \mathcal{F}^*_{g, \mathbb{P}^e}$, $h_c \circ g_c \in \mathcal{F}^*_{g_c}$, there is a set $\mathcal{B} = \{x \mid h(\phi(g(x))) \neq h(g_c(x))\} \neq \emptyset$.

We can construct undesirable target domains $\mathbb{P}^{e_i}$ with arbitrary loss $\mathcal{L}(h \circ g, \mathbb{P}^{e_i})$ by giving $(1 - \delta)$ percentage mass to that examples in $\mathcal{B}$ and $(\delta)$ percentage mass that examples in $\mathcal{X} \setminus \mathcal{B}$. This is equivalent to

$$\mathbb{E}_{(x,y) \sim \mathbb{P}^{e_i}} [h(g(x)) \neq h_c(g_c(x))] \geq 1 - \delta. \tag{11}$$

with $(0 \leq \delta \leq 1)$. $\qquad\square$

**Corollary A.8.** *(Corollary 4.1 in the main paper)* *Given* $g \in \mathcal{G}_s$*, there exists* $f = h \circ g \in \bigcap_{e \in \mathcal{E}_{train}} \mathcal{F}_{g, \mathbb{P}^e}$ *such that for any* $0 < \delta < 1$*, there are many undesirable target domains* $\mathbb{P}^T \sim \mathcal{P}$ *such that:*

$$\mathbb{E}_{(x,y) \sim \mathbb{P}^T} [f(x) \neq f^*(x)] \geq 1 - \delta.$$

*with* $f^* \in \mathcal{F}^*$*.*

*Proof.* Denote $\mathbb{P}^{\mathcal{E}_{tr}}$ is the mixture of training domains, then $\text{supp}\{\mathbb{P}^{\mathcal{E}_{tr}}(Z_c)\} = \cup_{e \in \mathcal{E}_{tr}} \text{supp}\{\mathbb{P}^e(Z_c)\} = \mathcal{Z}_c$. Additionally, given $g \in \mathcal{G}_s$, then there exists $\phi$ such that $g_c = \phi \circ g \in \mathcal{G}_c$.

Based on structural causal model (SCM) depicted in Figure 1, we have $Z_e \not\perp Y$ i.e., the environmental feature $Z_e$ spuriously correlated with $Y$. Hence, there exist $h \notin \{h_c \circ \phi \mid h_c \circ g_c \in \mathcal{F}_{g_c, \mathbb{P}^{\mathcal{E}_{tr}}}\}$ e.g., $h$ can rely on spurious feature $z_e$ (or both $z_c$ and $z_e$) to make predict for some $\{x \mid x = \psi_x\{z_c, z_e, u_x\}$ for some $z_c$ such that $\mathbb{P}(Y \mid_{z_e} = z_e) = \mathbb{P}(Y \mid z_c = z_c)\}$.

There is a set $\mathcal{B} = \{x \mid x = \psi_x\{z'_c, z_e, u_x\}$ for some $z'_c$ such that $\mathbb{P}(Y \mid_{z_e} = z_e) \neq \mathbb{P}(Y \mid z_c = z'_c)\} \neq \emptyset$. Consequently, $h(\phi(g(x))) \neq h_c(g_c(x))$ for all $x \in \mathcal{B}$

We can construct undesirable target domains $\mathbb{P}^{e_i}$ with arbitrary loss $\mathcal{L}(h \circ g, \mathbb{P}^{e_i})$ by giving $(1 - \delta)$ percentage mass to that examples in $\mathcal{B}$ and $(\delta)$ percentage mass that examples in $\mathcal{X} \setminus \mathcal{B}$. This is equivalent to

$$\mathbb{E}_{(x,y) \sim \mathbb{P}^{e_i}} [h(g(x)) \neq h_c(g_c(x))] \geq 1 - \delta.$$

with $(0 \leq \delta \leq 1)$.

By Theorem 3.6, $h_c \circ g_c \in \mathcal{F}_{g_c, \mathbb{P}^{\mathcal{E}_{tr}}}$ implies $h_c \circ g_c \in \mathcal{F}^*$. This concludes the proof.

$\qquad\square$

**Theorem A.9.** *(Theorem 3.4 in the main paper)* *Given sequence of training domains* $\mathcal{E}_{tr} = \{e_1, ..., e_K\} \subset \mathcal{E}$*, denote* $\mathcal{F}^k_\cap = \bigcap^k_{i=1} \mathcal{F}_{\mathbb{P}^{e_i}}$*. We consider* $\mathcal{E}_{tr}$ *to be **diverse** if for domain* $e_k$*, there exists at least one sample* $x = \psi_x(z_c, z_e, u_x)$ *such that* $\exists f \in \mathcal{F}^{k-1}_\cap : f(x) \neq \mathbb{P}(Y \mid z_c)$*. Given a set of diverse domains* $\mathcal{E}_{tr}$*, we have:*

$$\mathcal{F}^1_\cap \supset \mathcal{F}^2_\cap \supset ... \supset \mathcal{F}^K_\cap$$

*and the number of training domains $\mathcal{E}_{tr}$ is sufficiently large:*

$$\lim_{\mathcal{E}_{tr} \to \mathcal{E}} \mathcal{F}_{\cap}^{|\mathcal{E}_{tr}|} \to \mathcal{F}^*.$$

*Proof.* We prove the first statement by induction. Consider the case $\mathcal{F}_{k-1}^{\cap}$ and $\mathcal{F}_{k}^{\cap}$, we will show that if $\mathcal{E}_{tr}$ is considered as **diverse** $\mathcal{F}_{k-1}^{\cap} \supset \mathcal{F}_{k}^{\cap}$.

We have $\mathcal{F}_{k-1}^{\cap} \supseteq \mathcal{F}_{k}^{\cap}$ is obvious by definition. By definition of "diverse" training domains $\mathcal{E}_{tr}$, there exists at least one sample $x = \psi_x(z_c, z_e, u_x)$ such that $\exists f \in \mathcal{F}_{\cap}^{k-1} : f(x) \neq \mathbb{P}(Y \mid z_c)$. This means $f \notin \mathcal{F}_{k}^{\cap}$, hence, $\mathcal{F}_{k-1}^{\cap} \supset \mathcal{F}_{k}^{\cap}$.

For the second statement, we need to show that if $\mathcal{E}_{tr} = \mathcal{E}$ then $\mathcal{F}_{\cap}^{|\mathcal{E}_{tr}|} = \mathcal{F}^*$. This holds true by the definition of $\mathcal{F}^*$.

$\square$

## A.2 REPRESENTATION ALIGNMENT TRADE-OFF

As a reminder, $\mathbb{P}$ denotes data distribution on data space $\mathcal{X}$, while $g_{\#}\mathbb{P}$ denotes latent distribution on full latent space $\mathcal{Z}$, with $g : \mathcal{X} \mapsto \mathcal{Z}$ is the encoder.

In the following, we recap the theoretical results for Hellinger distance as presented by Phung et al. (2021). Similar results for $\mathcal{H}$-divergence can be found in Zhao et al. Zhao et al. (2019), and for Wasserstein distance in Le et al. Le et al. (2021).

### A.2.1 UPPER BOUND

**Theorem A.10.** *Consider the source domain $\mathbb{P}^{e'}$ and the target domain $\mathbb{P}^e$. Let $\ell$ be any loss function upper-bounded by a positive constant $L$. For any hypothesis $f : \mathcal{X} \mapsto \mathcal{Y}_\Delta$ where $f = h \circ g$ with $g : \mathcal{X} \mapsto \mathcal{Z}$ and $h : \mathcal{Z} \mapsto \mathcal{Y}_\Delta$, the target loss on input space is upper bounded*

$$\mathcal{L}\left(f, \mathbb{P}^e\right) \leq \mathcal{L}\left(f, \mathbb{P}^{e'}\right) + L\sqrt{2}\, d_{1/2}\left(\mathbb{P}_g^e, \mathbb{P}_g^{e'}\right), \tag{12}$$

This Theorem is directly adapted from the result of Trung et al. Phung et al. (2021). The upper bound for target loss above relates source loss, target loss and data shift on feature space, which is different to other bounds in which the data shift is on input space.

### A.2.2 LOWER BOUND

**Theorem A.11.** *Phung et al. (2021) Consider a hypothesis $f = h \circ g$, the Hellinger distance between two label marginal distributions $\mathbb{P}^{e'}$ and $\mathbb{P}^e$ can be upper-bounded as:*

$$d_{1/2}\left(\mathbb{P}_{\mathcal{Y}}^{e'}, \mathbb{P}_{\mathcal{Y}}^e\right) \leq \mathcal{L}\left(f, \mathbb{P}^{e'}\right)^{1/2} + d_{1/2}\left(g_{\#}\mathbb{P}^{e'}, g_{\#}\mathbb{P}^e\right) + \mathcal{L}\left(f, \mathbb{P}^e\right)^{1/2} \tag{13}$$

*where the general loss $\mathcal{L}$ is defined based on the Hellinger loss $\ell$ which is define as $\ell\left(f(x)\right) = D_{1/2}\left(f(x), \mathbb{P}(Y \mid x)\right) = 2 \sum_{i=1}^C \left(\sqrt{f(x,i)} - \sqrt{\mathbb{P}(Y = i \mid x)}\right)^2$.*

## A.3 SUBSPACE REPRESENTATION ALIGNMENT

In the following, we prove the theoretical results for Hellinger distance based on the findings of Trung et al. Phung et al. (2021). A similar strategy can be directly applied to $\mathcal{H}$-divergence Zhao et al. (2019) and Wasserstein distance Le et al. (2021).

**Theorem A.12.** *(Theorem 5.1 in the main paper) Given a subspace projector $\Gamma$, if the loss function $\ell$ is upper-bounded by a positive constant $L$, then:*

*(i) The subspace target general loss is upper-bounded:*

$$\frac{1}{|\mathcal{E}_{tr}|} \sum_{e \in \mathcal{E}_{tr}} \mathcal{L}\left(f, \mathbb{P}^e\right) \leq \sum_{e,e' \in \mathcal{E}_{tr}} \sum_{m \in \mathcal{M}} \pi_m^e \mathcal{L}\left(f, \mathbb{P}_m^{e'}\right) + \sum_{e,e' \in \mathcal{E}_{tr}} L\sqrt{2} \sum_{m \in \mathcal{M}} \pi_m^e d_{1/2}\left(g_{\#}\mathbb{P}_m^e, g_{\#}\mathbb{P}_m^{e'}\right),$$

*(ii) The Hellinger distance between two label marginal distributions on subspace $\mathbb{P}_{\mathcal{Y},m}^{e'}$ and $\mathbb{P}_{\mathcal{Y},m}^{e}$ can be upper-bounded:*

$$d_{1/2}\left(\mathbb{P}_{\mathcal{Y},m}^{e'}, \mathbb{P}_{\mathcal{Y},m}^{e}\right) \leq d_{1/2}\left(g_{\#}\mathbb{P}_m^{e'}, g_{\#}\mathbb{P}_m^{e}\right) + \mathcal{L}\left(f, \mathbb{P}_m^{e'}\right)^{1/2} + \mathcal{L}\left(f, \mathbb{P}_m^{e}\right)^{1/2}$$

*where $g_{\#}\mathbb{P}$ denotes representation distribution on representation space $\mathcal{Z}$ induce by applying encoder with $g : \mathcal{X} \mapsto \mathcal{Z}$ on data distribution $\mathbb{P}$, $D_{1/2}\left(\mathbb{P}^1(W), \mathbb{P}^2(W)\right) = 2\int\left(\sqrt{p^1(w)} - \sqrt{p^2(w)}\right)^2 dw$ is the Hellinger divergence Hellinger (1909) between two distributions. The squared $d_{1/2} = \sqrt{D_{1/2}}$ is a proper metric, the general loss $\mathcal{L}$ is defined based on the Hellinger loss $\ell$ which is define as $\ell\left(f(x)\right) = D_{1/2}\left(f(x), \mathbb{P}\left(Y \mid x\right)\right) = 2\sum_{i=1}^{C}\left(\sqrt{f(x,i)} - \sqrt{\mathbb{P}\left(Y = i \mid x\right)}\right)^2.$*

*Proof.* We consider *sub-space projector* $\Gamma : \mathcal{X} \to \mathcal{M}$, given a sub-space index $m \in \mathcal{M}$, we denote $A_m = \Gamma^{-1}(m) = \{x : \Gamma(x) = m\}$ is the region on data space which has the same index $m$. Let $\mathbb{P}_m^e$ be the distribution restricted by $\mathbb{P}^e$ over the set $A_m$ and $\mathbb{P}_m^e$ as the distribution restricted by $\mathbb{P}^e$ over $A_m$. Eventually, we define $\mathbb{P}_m^e\left(y \mid x\right)$ as the probabilistic labeling distribution on the sub-space $(A_m, \mathbb{P}_m^e)$, meaning that if $x \sim \mathbb{P}_m^e$, $\mathbb{P}_m^e\left(y \mid x\right) = \mathbb{P}_e\left(y \mid x\right)$. Similarly, we define if $x \sim \mathbb{P}_m^{e'}$, $\mathbb{P}_m^{e'}\left(y \mid x\right) = \mathbb{P}^{e'}\left(y \mid x\right)$. Due to this construction, any data sampled from $\mathbb{P}_m^e$ or $\mathbb{P}_m^{e'}$ have the same index $m = \Gamma(x)$. Additionally, since each data point $x \in \mathcal{X}$ corresponds to only a single $\Gamma(x)$, the data space is partitioned into disjoint sets, i.e., $\mathcal{X} = \bigcup_{m=1}^{\mathcal{M}} A_m$, where $A_m \cap A_n = \emptyset, \forall m \neq n$. Consequently, the general loss of the target domain becomes:

$$\mathcal{L}\left(f, \mathbb{P}^e\right) := \sum_{m \in \mathcal{M}} \pi_m^e \mathcal{L}\left(f, \mathbb{P}_m^e\right), \tag{14}$$

where $\mathcal{M}$ is the set of all feasible sub-spaces indexing $m$ and $\pi_m^e = \frac{\mathbb{P}^e(A_m)}{\sum_{m' \in \mathcal{M}} \mathbb{P}^e(A_{m'})}$.

We obtain point (i) directly by applying the results from Theorem A.11 to each individual sub-space, denoted by the index $m$.

Using the same proof for a single space in Theorem A.10, we obtain:

$$\mathcal{L}\left(f, \mathbb{P}_m^e\right) \leq \mathcal{L}\left(f_m, \mathbb{P}_m^{e'}\right) + L\sqrt{2}d_{1/2}\left(g_{\#}\mathbb{P}_m^e, g_{\#}\mathbb{P}_m^{e'}\right) \tag{15}$$

Since $\mathcal{L}\left(f, \mathbb{P}^e\right) := \sum_m \pi_m^e \mathcal{L}\left(f, \mathbb{D}_m^e\right)$, taking weighted average over $m \in \mathcal{M}$, we reach (ii):

$$\mathcal{L}\left(f, \mathbb{P}^e\right) \leq \sum_m \pi_m^e \mathcal{L}\left(f_m, \mathbb{P}_m^{e'}\right) + L\sqrt{2} \sum_m \pi_m^e d_{1/2}\left(g_{\#}\mathbb{P}_m^e, g_{\#}\mathbb{P}_m^{e'}\right) \tag{16}$$

By summing over the training domains on the left-hand side, we obtain:

$$\sum_{e \in \mathcal{E}_{tr}} \mathcal{L}\left(f_{\mathcal{M}}, \mathbb{P}^e\right) \leq \sum_{e \in \mathcal{E}_{tr}} \sum_m \pi_m^e \mathcal{L}\left(f_m, \mathbb{P}_m^{e'}\right) + \sum_{e \in \mathcal{E}_{tr}} L\sqrt{2} \sum_m \pi_m^e d_{1/2}\left(g_{\#}\mathbb{P}_m^e, g_{\#}\mathbb{P}_m^{e'}\right)$$

Summing over the training domains on the left-hand side again:

$$\sum_{e' \in \mathcal{E}_{tr}} \sum_{e \in \mathcal{E}_{tr}} \mathcal{L}\left(f_{\mathcal{M}}, \mathbb{P}^e\right) \leq \sum_{e' \in \mathcal{E}_{tr}} \sum_{e \in \mathcal{E}_{tr}} \sum_m \pi_m^e \mathcal{L}\left(f_m, \mathbb{P}_m^{e'}\right)$$
$$+ \sum_{e' \in \mathcal{E}_{tr}} \sum_{e \in \mathcal{E}_{tr}} L\sqrt{2} \sum_m \pi_m^e d_{1/2}\left(g_{\#}\mathbb{P}_m^e, g_{\#}\mathbb{P}_m^{e'}\right)$$

Finally, we obtain:

$$|\mathcal{E}_{tr}| \sum_{e \in \mathcal{E}_{tr}} \mathcal{L}\left(f, \mathbb{P}^e\right) \leq \sum_{e,e' \in \mathcal{E}_{tr}} \sum_{m \in \mathcal{M}} \pi_m^e \mathcal{L}\left(f, \mathbb{P}_m^{e'}\right) + \sum_{e,e' \in \mathcal{E}_{tr}} L\sqrt{2} \sum_{m \in \mathcal{M}} \pi_m^e d_{1/2}\left(g_{\#}\mathbb{P}_m^e, g_{\#}\mathbb{P}_m^{e'}\right)$$
$$\tag{17}$$

$\square$

# B  ADDITIONAL DISCUSSION WITH RELATED WORKS

**Optimal Representation (Ruan et al., 2021)**  While at first glance, (Ruan et al., 2021) and our work share the same goal of identifying the necessary and sufficient conditions for generalization, the two studies fundamentally differ in the following aspects:

Ruan et al. (2021) aim to identify the set of conditions that are both necessary and sufficient, which provide theoretical guarantee essentially by assuming some knowledge of target domains. Without accessing target information, generalization is provably impossible. Meanwhile, we focus on analyzing generalizability from limited domains without assuming any additional information from the target.

More concretely, Ruan et al. (2021) propose the *idealized* domain generalization hypothesis (IDG), which is the expected worst-case target risk over source risk minimizers:

$$R_{IDG} = \mathbb{E}_{e_i, e_j \sim \mathcal{P}} \left[ \sup_{f \in \mathcal{F}_{\mathbb{P}^{e_i}}} \mathcal{L}(f, \mathbb{P}^{e_i}) \right]$$

$R_{IDG}$ is an expectation over all possible pairs of domains $(e_i, e_j) \sim \mathcal{P}$ where $\mathcal{P}$ is the distribution over domain space $\mathcal{E}$. During training, they sample any two domains from the domain distribution, assigning one as the source and the other as the target, to determine the worst-case target risk.

The representation $Z = g(X)$ deemed optimal for IDG must satisfy two conditions (by Theorem 1 therein):

- Sufficient representation: the representation needs to be task-discriminative, allowing a predictor to minimize risk across all domains. In the presence of all domains, this condition can be simply satisfied by learning a hypothesis optimal for all training domains.
- The representation's marginal support must be consistent across all pairs of source and target domains. This condition generally coincides with our assumption of causal support, which is a common assumption across DG literature.

It is clear from the formulation $R_{IDG}$ that *all* possible domains should be known to achieve generalization. Ruan et al. (2021) also point out the challenge in generalization without data from the target domain and recommends incorporating data augmentation from pre-trained models such as CLIP. To our best knowledge, using augmentation in DG is not new. Various studies have shown that access to all label-preserving augmentations (which is generally unfeasible) would reveal true causal factors (Mitrovic et al., 2020; Gao et al., 2023). To satisfy this condition, Ruan et al. (2021) assume augmentation is Bayes-preserving augmentation (Assumption 10 therein).

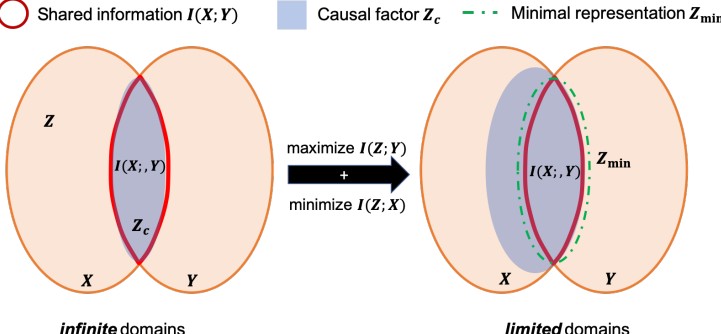

Figure 4: Information diagrams of $X, Y, Z_c$ and $Z_{\min} := g(X)$ s.t $g \in \{\arg\max_g I(g(X); Y) - I(g(X); X)\}$. In limited training domains, learning such minimal representation $Z_{\min}$ would capture the least information about $Z_c$.

**Information Bottleneck Theory.** We here elucidate our claim in Section 4.2 that minimizing $I(g(X); X)$ can subject the model to violating *Condition* 3.7. Whereas Ahuja et al. (2021) posits that information bottleneck aids generalization, such methods in fact assume sufficient and diverse domains, that is when a sufficient condition is met. In this case, the information about $Z_c$ is fully covered by the region $I(X; Y)$ and any $g = \arg\min_g I(g(X); X)$ s.t $g \in \{\arg\max_g I(g(X); Y)\}$ could guarantee all spurious features are discarded.

When the training domains are limited, the learned representations is however more likely to contain spurious correlations bad for prediction on unseen domains. Thus, minimizing $I(g(X); X)$ in fact would at most capture the shared information of $X$ and $Y$, thus yielding representations with the least information about $Z_c$. Therefore, such minimal representations are the least likely to meet the sufficient representation constraint in practice. Figure 4 illustrates the difference between two learning scenarios.

# C  PRACTICAL METHODOLOGY

In this section, we present the practical objectives to achieve Eq. (4):

$$\min_f \underbrace{\sum_{e, e' \in \mathcal{E}_{tr}} \sum_{m \in \mathcal{M}} D\left(g\#\mathbb{P}_m^e, g\#\mathbb{P}_m^{e'}\right)}_{\text{Subspace Representation Alignment}} \text{ s.t. } f \in \underbrace{\bigcap_{e \in \mathcal{E}_{tr}} \arg\min_f \mathcal{L}\left(f, \mathbb{P}^e\right)}_{\text{Training domain optimal hypothesis}} \tag{18}$$

where $\mathcal{M} = \{\hat{y} \mid \hat{y} = f(x), x \in \bigcup_{e \in \mathcal{E}_{tr}} \text{supp}\mathbb{P}^e\}$ and $D$ can be $\mathcal{H}$-divergence, Hellinger distance, Wasserstein distance.

In the following, we consider the encoder $g$, classifier $h$, domain discriminator $h_d$ and set of $K$ empirical training domains $\mathbb{D}^{e_i} = \{x_j^{e_i}, y_j^{e_i}\}_{j=1}^{N_{e_i}} \sim [\mathbb{P}^{e_i}]^{N_{e_i}}, i = 1...K$.

## C.1  OPTIMAL HYPOTHESIS ACROSS TRAINING DOMAINS

For *optimal hypothesis across training domains condition*, we simply adopting the objective set forth by ERM:

$$\min_f \sum_{i=1}^{K} \mathcal{L}\left(f, \mathbb{D}^{e_i}\right) \tag{19}$$

## C.2  SUBSPACE REPRESENTATION ALIGNMENT

**Subspace Modelling and Projection.** Our objective is to map samples $x$ from training domains with identical predictions $f(x) = m$ into a unified subspace, where $m \in \mathcal{M} = \{\hat{y} \mid \hat{y} = f(x), x \in \bigcup_{e \in \mathcal{E}_{tr}} \text{supp}\mathbb{P}^e\}$. Given that the cardinality of $\mathcal{M}$ can be exceedingly large, potentially equal to the total number of training samples if the output of $f(x)$ is unique for each sample, this makes the optimization process particularly challenging.

Drawing inspiration from the concept of prototypes Snell et al. (2017), we suggest representing $\mathcal{M}$ as a set of prototypes $\mathcal{M} = \{m_i\}_{i=1}^{M}$, where each $m_i$ is an element of $\mathcal{Z}$. Consequently, a sample $x$ is assigned to a subspace by selecting the nearest prototype $m_i$ i.e., $i = \arg\min_{i'} \rho(g(x), m_{i'})$. Note that prototypes act as condensed representations of specific prediction outcomes. Consequently, samples assigned to the same prototype will receive the same prediction. Although this approach streamlines the subspace projection, it may lead to local optima as the mapping might favor a limited number of prototypes early in training Vuong et al. (2023). To mitigate this issue, we adopt a Wasserstein (WS) clustering approach Vuong et al. (2023) to guide the mapping of latent features from each domain into the designated subspace more effectively. We first endow a discrete distribution over the prototypes as $\mathbb{P}_{\mathcal{M}, \pi} = \sum_{i=1}^{M} \pi_i \delta_{m_i}$ with the Dirac delta function $\delta$ and the weights $\pi \in \Delta_M = \{\pi' \geq \mathbf{0} : \|\pi'\|_1 = 1\}$.

Then we project each domain $\mathbb{P}^{e_i}$ in subspaces indexed by prototypes as follows:

$$\min_{\mathcal{M},\pi} \min_{g} \left\{ \mathcal{L}_P = \sum_{i=1}^{K} \lambda \mathcal{W}_\rho \left( g \# \mathbb{P}^{e_i}, \mathbb{P}_{\mathcal{M},\pi} \right) \right\}, \tag{20}$$

where:

- Cost metric $\rho(z, m) = \frac{z^\top m}{\|z\|\|c\|}$ is the cosine similarity between the latent representation $z$ and the prototype $c$.

- Wasserstein distance between source domain representation distribution and distribution over prototype $\mathbb{P}_{\mathcal{M},\pi}$:

$$\mathcal{W}_d \left( g \# \mathbb{P}_x^{e_i}, \mathbb{P}_{c,\pi} \right) = \mathcal{W}_d \left( \sum_{n=1}^{B} \frac{1}{B} g(x_n), \sum_{i=1}^{M} \pi_i \delta_{m_i} \right) \tag{21}$$

$$= \frac{1}{B} \min_{\Gamma : \Gamma \# \left( g \# \mathbb{P}_x^{e_i} \right) = \mathbb{P}_{c,\pi}} \sum_{n=1}^{B} \rho \left( g(x_n), \Gamma(g(x_n)) \right) \tag{22}$$

Where $B$ is the batch size. This Wasserstein distance can be effectively compute by linear dynamic programming method, entropic dual form of optimal transport (Genevay et al., 2016) or Sinkhorn algorithm Cuturi (2013).

**Subspace Alignment Constraints** Subspace alignment is achieved through a conditional adversarial training approach Gan et al. (2016); Li et al. (2018b). In this framework, the **subspace-conditional** domain discriminator $h_d$ aims to accurately predict the domain label "$e_i$" based on the combined feature $[z, m]$, where $\{z = g(x), m = \Gamma(x)\}$. Concurrently, the objective for the representation function $g$ is to transform the input $x$ into a latent representation $z$ in such a way that $h_d$ is unable to determine the domain "$e_i$" of $x$. We employ the Gradient Reversal Layer (GRL) as introduced byGanin et al. (2016), thereby simplifying the optimization process to:

$$\min_{g,h_d} \left\{ \mathcal{L}_D = - \sum_{i=1}^{K} \mathbb{E}_{x \sim \mathbb{D}^{e_i}} \left[ e_i \log h_d \left( [\mathcal{R}(g(x)), m] \right) \right] \right\} \tag{23}$$

where $\mathcal{R}$ is gradient reversal layer.

FINAL OBJECTIVE

Putting all together, we propose a joint optimization objective, which is given as

$$\min_{\mathcal{M},\pi} \min_{g,h,h_d} \left\{ \mathcal{L}_H + \lambda_P \mathcal{L}_P + \lambda_D \mathcal{L}_D \right\}, \tag{24}$$

where $\lambda_P, \lambda_D$ are regularization hyperparameters.

C.3 ABLATION STUDY ON THE NUMBER OF SUBSPACES

Considering our data generation process, the number of distinct labels $\mathbb{P}(Y \mid x)$ reflects the number of distinct causal factors (denoted as $|\mathcal{Z}|$). If $\mathcal{M} \leq |\mathcal{Z}|$, samples with different labels may be projected into the same subspace, leading to discrepancies in the marginal label distribution within that subspace.

We revisit the two key points in the previous discussion:

- Theorem 5.1 implies that projecting samples into the correct subspaces can significantly reduce or entirely eliminate marginal label shifts within those subspaces, assuming optimal projection for the sake of simplicity.
- As mentioned earlier, projecting samples with the same label $\mathbb{P}(Y \mid x)$ eliminates the discrepancy $d_{1/2} \left( \mathbb{P}_{\mathcal{Y},m}^{e_i}, \mathbb{P}_{\mathcal{Y},m}^{e_i} \right)$, reducing it to zero.

Increasing $\mathcal{M}$ reduces the likelihood of differently labeled samples being mapped to the same subspace, thus decreasing the discrepancy outlined in Theorem 5.1 (ii). It's notable that the upper bound in (i) can be optimized to the limit defined by (ii) when the focus is only on training domains. This optimization, in turn, minimizes the bound (i).

Rather than treating $|\mathcal{Z}|$ merely as a parameter for tuning, we delve further into analyzing the impact of varying $|\mathcal{Z}|$ values. In this ablation study, we test $|\mathcal{Z}|$ values of $[4, 8, 16, 32] \times |\mathcal{C}|$, where $|\mathcal{C}|$ denotes the number of classes.

Table 3: Classification Accuracy on PACS using ResNet50 with different number of subspaces (NoS) per class.

| NoS $|\mathcal{M}|$ | A | C | P | S | Avg |
|---|---|---|---|---|---|
| ERM | $89.3 \pm 0.2$ | $83.4 \pm 0.6$ | $97.3 \pm 0.3$ | $82.5 \pm 0.5$ | 88.1 |
| 4 | $90.2 \pm 0.3$ | $83.2 \pm 0.7$ | $97.9 \pm 0.2$ | $82.3 \pm 1.5$ | 88.2 |
| 8 | $90.5 \pm 0.8$ | $83.8 \pm 0.6$ | $97.6 \pm 0.3$ | $82.1 \pm 1.8$ | 88.7 |
| 16 | $90.5 \pm 0.5$ | $83.4 \pm 0.2$ | $97.8 \pm 0.1$ | $83.2 \pm 0.2$ | 88.7 |
| 32 | $90.2 \pm 0.5$ | $83.8 \pm 0.8$ | $97.3 \pm 0.4$ | $82.0 \pm 1.2$ | 88.4 |

Table 3 reveals that performance generally improves with an increase in the number of prototypes. Nonetheless, a decline in performance is noted when $K$ becomes excessively large. We speculate this behavior is tied to the dataset's underlying causal factors; specifically, if a limited number of causal factors generate the data, assigning a large number of prototypes to capture discriminative information might result in one causal factor being associated with multiple prototypes, thereby introducing ambiguity. This hypothesis, however, requires further investigation for confirmation, and we earmark it for future research.

### C.4 COMPARE TO OTHER BASELINES

One of our key contributions is offering a new perspective on why domain generalization (DG) algorithms often fail to outperform the fundamental empirical risk minimization (ERM) approach on standard benchmarks, through an analysis of sufficient and necessary conditions. In the main paper, we compare our proposed SRA method with the two most related methods, DANN and CDANN, as they represent specific cases of our approach where the number of subspaces per class is set to 0 and 1, respectively.

In this section, we provide additional experimental results from various baselines, both with and without SWAD, on five datasets from the DomainBed benchmark Gulrajani & Lopez-Paz (2021), to further support our discussion and analysis.

Table 4: Classification accuracy (%) for all algorithms across datasets.

| Algorithm | VLCS | PACS | OfficeHome | TerraIncognita | DomainNet | Avg |
|---|---|---|---|---|---|---|
| ERM (Gulrajani & Lopez-Paz, 2021) | $77.5 \pm 0.4$ | $85.5 \pm 0.2$ | $66.5 \pm 0.3$ | $46.1 \pm 1.8$ | $40.9 \pm 0.1$ | 63.3 |
| GroupDRO (Sagawa et al., 2019) | $76.7 \pm 0.6$ | $84.4 \pm 0.8$ | $66.0 \pm 0.7$ | $43.2 \pm 1.1$ | $33.3 \pm 0.2$ | 60.7 |
| Mixup (Wang et al., 2020b) | $77.4 \pm 0.6$ | $84.6 \pm 0.6$ | $68.1 \pm 0.3$ | $47.9 \pm 0.8$ | $39.2 \pm 0.1$ | 63.4 |
| MLDG (Li et al., 2018a) | $77.2 \pm 0.4$ | $84.9 \pm 1.0$ | $\mathbf{66.8} \pm 0.6$ | $47.7 \pm 0.9$ | $41.2 \pm 0.1$ | 63.6 |
| MTL (Blanchard et al., 2021) | $77.2 \pm 0.4$ | $84.6 \pm 0.5$ | $66.4 \pm 0.5$ | $45.6 \pm 1.2$ | $40.6 \pm 0.1$ | 62.9 |
| SagNet (Nam et al., 2021) | $77.8 \pm 0.5$ | $\mathbf{86.3} \pm 0.2$ | $68.1 \pm 0.1$ | $48.6 \pm 1.0$ | $40.3 \pm 0.1$ | 64.2 |
| ARM (Zhang et al., 2021) | $76.6 \pm 0.3$ | $85.1 \pm 0.4$ | $64.8 \pm 0.3$ | $45.5 \pm 0.3$ | $35.5 \pm 0.2$ | 61.7 |
| RSC (Huang et al., 2020) | $77.1 \pm 0.5$ | $85.2 \pm 0.9$ | $65.5 \pm 0.9$ | $46.6 \pm 1.0$ | $38.9 \pm 0.5$ | 62.7 |
| IRM (Arjovsky et al., 2020) | $78.5 \pm 0.5$ | $83.5 \pm 0.8$ | $64.3 \pm 2.2$ | $47.6 \pm 0.8$ | $33.9 \pm 2.8$ | 61.6 |
| MMD (Li et al., 2018b) | $77.5 \pm 0.9$ | $84.6 \pm 0.5$ | $66.3 \pm 0.1$ | $42.2 \pm 1.6$ | $23.4 \pm 9.5$ | 58.8 |
| CORAL (Sun & Saenko, 2016) | $\mathbf{78.8} \pm 0.6$ | $86.2 \pm 0.3$ | $\mathbf{68.7} \pm 0.3$ | $47.6 \pm 1.0$ | $41.5 \pm 0.1$ | 64.5 |
| VREx (Krueger et al., 2021) | $78.3 \pm 0.2$ | $84.9 \pm 0.6$ | $66.4 \pm 0.6$ | $46.4 \pm 0.6$ | $33.6 \pm 2.9$ | 61.9 |
| DANN (Ganin et al., 2016) | $78.6 \pm 0.4$ | $83.6 \pm 0.4$ | $65.9 \pm 0.6$ | $46.7 \pm 0.5$ | $38.3 \pm 0.1$ | 62.6 |
| CDANN (Li et al., 2018b) | $77.5 \pm 0.1$ | $82.6 \pm 0.9$ | $65.8 \pm 1.3$ | $45.8 \pm 1.6$ | $38.3 \pm 0.3$ | 62.0 |
| **Ours** (SRA) | $76.4 \pm 0.7$ | $\mathbf{86.3} \pm 1.1$ | $66.4 \pm 0.7$ | $\mathbf{49.5} \pm 1.0$ | $\mathbf{44.5} \pm 0.3$ | **64.6** |

As observed in both Table 4 and Table 5, the baselines fail to consistently surpass the simple ERM baseline across all settings. While some methods perform well on certain datasets, they perform worse on others. However, the combination of our proposed method (SRA), which enforces strong sufficient conditions, and SWAD, which promotes necessary conditions, significantly improves generalization. This combination outperforms ERM and other baselines in all settings. These results support our analysis in Section 4.2, indicating that existing methods often violate the necessary condition for effective domain generalization.

Table 5: Classification accuracy (%) for all algorithms across datasets.

| Algorithm | VLCS | PACS | OfficeHome | TerraIncognita | Avg |
|---|---|---|---|---|---|
| SWAD (Cha et al., 2021) | $79.1 \pm 0.4$ | $88.1 \pm 0.4$ | $70.6 \pm 0.3$ | $50.0 \pm 0.4$ | 72.0 |
| SWAD + IRM (Arjovsky et al., 2020) | $78.8 \pm 0.2$ | $88.1 \pm 0.4$ | $70.4 \pm 0.2$ | $49.6 \pm 1.7$ | 71.7 |
| SWAD + VREx (Krueger et al., 2021) | $78.1 \pm 1.3$ | $85.4 \pm 0.5$ | $69.9 \pm 0.1$ | $50.0 \pm 0.2$ | 70.9 |
| SWAD +CORAL (Sun & Saenko, 2016) | $\underline{78.9} \pm 0.6$ | $88.3 \pm 0.5$ | $71.4 \pm 0.1$ | $51.1 \pm 0.9$ | 72.4 |
| SWAD +MMD (Li et al., 2018b) | $78.7 \pm 0.1$ | $88.3 \pm 0.1$ | $70.6 \pm 0.4$ | $49.6 \pm 0.5$ | 71.8 |
| SWAD + DANN | $79.2 \pm 0.0$ | $87.9 \pm 0.5$ | $70.5 \pm 0.1$ | $50.6 \pm 0.6$ | 72.2 |
| SWAD + CDANN | $79.3 \pm 0.2$ | $87.7 \pm 0.3$ | $70.4 \pm 0.1$ | $50.7 \pm 0.1$ | 72.2 |
| **Ours** (SRA + SWAD) | $\underline{79.4} \pm 0.4$ | $\mathbf{88.7} \pm 0.2$ | $72.1 \pm 0.5$ | $\underline{51.6} \pm 1.2$ | $\underline{73.0}$ |
| **Ours** (SRA + SWAD + Ensemble) | $\mathbf{79.8} \pm 0.0$ | $\mathbf{89.2} \pm 0.0$ | $\mathbf{73.2} \pm 0.0$ | $\mathbf{52.2} \pm 0.0$ | $\mathbf{73.3}$ |

# D  FULL EXPERIMENTAL RESULTS

**Metrics.**  We adopt the training and evaluation protocol as in DomainBed benchmark (Gulrajani & Lopez-Paz, 2021), including dataset splits, hyperparameter (HP) search, model selection on the validation set, and optimizer HP. To manage computational demands more efficiently, as suggested by (Cha et al., 2021), we narrow our HP search space. Specifically, we use the Adam optimizer, as detailed in (Gulrajani & Lopez-Paz, 2021), setting the learning rate to a default of $5e^{-5}$ and forgoing dropout and weight decay adjustments. The batch size is maintained at 32. For DomainNet, we run a total of 15,000 iterations, while for other datasets, we limit iterations to 5,000, deemed adequate for model convergence. Our method's unique parameters, including the regularization hyperparameters $(\lambda_P, \lambda_D)$, undergo optimization within the range of $[0.01, 0.1, 1.0]$, and the number of prototypes $|\mathcal{Z}|$ is fixed at 16 times the number of classes. It is worth noting that while we conduct ablation study on PACS dataset, we utilize the number of prototypes $|\mathcal{Z}|$ is fixed at 16 times the number of classes for all datasets. SWAD-specific hyperparameters remain unaltered from their default settings. The evaluation frequency is set to 300 for all dataset.

Our code is anonymously published at `https://anonymous.4open.science/r/submisson-FCF0`.

## D.1  DATASETS

To evaluate the effectiveness of the proposed method, we utilize five datasets: PACS (Li et al., 2017), VLCS (Torralba & Efros, 2011), Office-Home (Venkateswara et al., 2017), Terra Incognita (Beery et al., 2018) and DomainNet (Peng et al., 2019) which are the common DG benchmarks with multi-source domains.

- **PACS** (Li et al., 2017):  9991 images of seven classes in total, over four domains:Art_painting (A), Cartoon (C), Sketches (S), and Photo (P).
- **VLCS** (Torralba & Efros, 2011):  five classes over four domains with a total of 10729 samples. The domains are defined by four image origins, i.e., images were taken from the PASCAL VOC 2007 (V), LabelMe (L), Caltech (C) and Sun (S) datasets.
- **Office-Home** (Venkateswara et al., 2017): 65 categories of 15,500 daily objects from 4 domains: Art, Clipart, Product (vendor website with white-background) and Real-World (real-object collected from regular cameras).
- **Terra Incognita** (Beery et al., 2018) includes 24,788 wild photographs of dimension (3, 224, 224) with 10 animals, over 4 camera-trap domains L100, L38, L43 and L46. This dataset contains photographs of wild animals taken by camera traps; camera trap locations are different across domains.
- **DomainNet** (Peng et al., 2019) contains 596,006 images of dimension (3, 224, 224) and 345 classes, over 6 domains clipart, infograph, painting, quickdraw, real and sketch. This is the biggest dataset in terms of the number of samples and classes.

## D.2  RESULTS

In this section, we present the extended results of Table 2 in the main text. The following tables report the domain-specific performance of each method on 5 datasets: VLCS (Table 6), PACS (Table 7), OfficeHome (Table 8), TerraIncognita (Table 9) and Domain Net (Table 10).

Standard errors are computed over three trials. Our models are run on 4 RTX 6000 GPU cores of 32GB. One full training routine takes roughly 2 hours.

Table 6: Classification Accuracy on **VLCS** using ResNet50

| Algorithm | C | L | S | V | Avg |
|---|---|---|---|---|---|
| ERM (Zhang et al., 2020) | 97.7 ± 0.4 | 64.3 ± 0.9 | 73.4 ± 0.5 | 74.6 ± 1.3 | 77.5 |
| DANN (Ganin et al., 2016) | 99.0 ± 0.3 | **65.1** ± 1.4 | 73.1 ± 0.3 | 77.2 ± 0.6 | 78.6 |
| CDANN (Li et al., 2018b) | 97.1 ± 0.3 | **65.1** ± 1.2 | 70.7 ± 0.8 | 77.1 ± 1.5 | 77.5 |
| **Ours** (SRA) | 97.1 ± 1.5 | 63.8 ± 2.3 | 70.5 ± 2.2 | 74.1 ± 1.8 | 76.4 |
| SWAD Cha et al. (2021) | 98.8 ± 0.1 | 63.3 ± 0.3 | 75.3 ± 0.5 | 79.2 ± 0.6 | 79.1 |
| SWAD + DANN | 99.2 ± 0.1 | 63.0 ± 0.8 | 75.3 ± 1.8 | 79.3 ± 0.5 | 79.2 |
| SWAD + CDANN | 99.1 ± 0.1 | 63.3 ± 0.7 | 75.1 ± 0.7 | 80.1 ± 0.2 | 79.3 |
| **Ours** (SRA + SWAD) | 98.9 ± 0.2 | 63.7 ± 0.3 | 75.6 ± 0.4 | 79.4 ± 0.8 | 79.4 |
| **Ours** (SRA + SWAD + Ensemble) | **99.1** ± 0.0 | 63.9 ± 0.0 | **76.3** ± 0.0 | **79.9** ± 0.8 | **79.8** |

Table 7: Classification Accuracy on **PACS** using ResNet50

| Algorithm | A | C | P | S | Avg |
|---|---|---|---|---|---|
| ERM (Gulrajani & Lopez-Paz, 2021) | 84.7 ± 0.4 | 80.8 ± 0.6 | 97.2 ± 0.3 | 79.3 ± 1.0 | 85.5 |
| DANN (Ganin et al., 2016) | 86.4 ± 0.8 | 77.4 ± 0.8 | 97.3 ± 0.4 | 73.5 ± 2.3 | 83.6 |
| CDANN (Li et al., 2018b) | 84.6 ± 1.8 | 75.5 ± 0.9 | 96.8 ± 0.3 | 73.5 ± 0.6 | 82.6 |
| **Ours** (SRA) | 86.4 ± 0.2 | 82.0 ± 0.8 | 96.7 ± 1.1 | 80.2 ± 4.4 | 86.3 |
| SWAD Cha et al. (2021) | 89.3 ± 0.2 | 83.4 ± 0.6 | 97.3 ± 0.3 | 82.5 ± 0.5 | 88.1 |
| SWAD + DANN | 90.7 ± 1.2 | 82.2 ± 0.4 | 97.3 ± 0.1 | 81.6 ± 0.4 | 87.9 |
| SWAD + CDANN | 90.5 ± 0.3 | 82.4 ± 1.0 | 97.6 ± 0.1 | 80.4 ± 0.3 | 87.7 |
| **Ours** (SRA + SWAD) | 90.5 ± 0.5 | 83.4 ± 0.2 | 97.8 ± 0.1 | 83.2 ± 0.2 | 88.7 |
| **Ours** (SRA + SWAD + Ensemble) | **91.2** ± 0.0 | **83.8** ± 0.0 | 97.8 ± 0.0 | **83.9** ± 0.0 | **89.2** |

Table 8: Classification Accuracy on **OfficeHome** using ResNet50

| Algorithm | A | C | P | R | Avg |
|---|---|---|---|---|---|
| ERM (Gulrajani & Lopez-Paz, 2021) | 61.3 ± 0.7 | 52.4 ± 0.3 | 75.8 ± 0.1 | 76.6 ± 0.3 | 66.5 |
| DANN (Ganin et al., 2016) | 59.9 ± 1.3 | 53.0 ± 0.3 | 73.6 ± 0.7 | 76.9 ± 0.5 | 65.9 |
| CDANN (Li et al., 2018b) | 61.5 ± 1.4 | 50.4 ± 2.4 | 74.4 ± 0.9 | 76.6 ± 0.8 | 65.8 |
| **Ours** (SRA) | 62.2 ± 1.4 | 52.3 ± 1.7 | 74.5 ± 0.8 | 76.6 ± 1.3 | 66.4 |
| SWAD Cha et al. (2021) | 66.1 ± 0.4 | 57.7 ± 0.4 | 78.4 ±0.1 | 80.2 ± 0.2 | 70.6 |
| SWAD + DANN | 67.2 ± 0.1 | 56.2 ± 0.1 | 78.6 ±0.2 | 80.0 ± 0.5 | 70.5 |
| SWAD + CDANN | 66.8 ± 0.4 | 56.4 ± 0.8 | 78.4 ±0.5 | 80.1 ± 0.2 | 70.4 |
| **Ours** (SRA + SWAD) | 69.1 ± 0.6 | 58.4 ± 0.8 | 79.5 ± 0.2 | 81.4 ± 0.3 | 72.1 |
| **Ours** (SRA + SWAD + Ensemble) | **70.5** ± 0.0 | **59.5** ± 0.0 | **80.4** ± 0.0 | **82.1** ± 0.0 | **73.2** |

Table 9: Classification Accuracy on **TerraIncognita** using ResNet50

| Algorithm | L100 | L38 | L43 | L46 | Avg |
|---|---|---|---|---|---|
| ERM (Gulrajani & Lopez-Paz, 2021) | 49.8 ± 4.4 | 42.1 ± 1.4 | 56.9 ± 1.8 | 35.7 ± 3.9 | 46.1 |
| DANN (Ganin et al., 2016) | 51.1 ± 3.5 | 40.6 ± 0.6 | 57.4 ± 0.5 | 37.7 ± 1.8 | 46.7 |
| CDANN (Li et al., 2018b) | 47.0 ± 1.9 | 41.3 ± 4.8 | 54.9 ± 1.7 | 39.8 ± 2.3 | 45.8 |
| **Ours** (SRA) | 52.9 ± 3.5 | 45.8 ± 5.1 | 57.2 ± 4.6 | 42.3 ± 1.1 | 49.5 |
| SWAD Cha et al. (2021) | 55.4 ± 0.0 | 44.9 ± 1.1 | 59.7 ± 0.4 | 39.9 ± 0.2 | 50.0 |
| SWAD + DANN | 56.3 ± 2.6 | 44.9 ± 0.4 | 60.0 ± 0.7 | 41.4 ± 0.3 | 50.6 |
| SWAD + CDANN | 55.2 ± 2.2 | 45.3 ± 0.2 | 61.4 ± 0.7 | 40.9 ± 2.0 | 50.7 |
| **Ours** (SRA + SWAD) | 56.2 ± 0.8 | 45.5 ± 2.6 | 60.4 ± 1.0 | 44.4 ± 0.6 | 51.6 |
| **Ours** (SRA + SWAD + Ensemble) | **57.4** ± 0.0 | **45.3** ± 0.0 | **60.9** ± 0.0 | **45.2** ± 0.0 | **52.2** |

Table 10: Classification Accuracy on **DomainNet** using ResNet50

| Algorithm | clip | info | paint | quick | real | sketch | Avg |
|---|---|---|---|---|---|---|---|
| ERM (Gulrajani & Lopez-Paz, 2021) | $58.1 \pm 0.3$ | $18.8 \pm 0.3$ | $46.7 \pm 0.3$ | $12.2 \pm 0.4$ | $59.6 \pm 0.1$ | $49.8 \pm 0.4$ | 40.9 |
| DANN (Ganin et al., 2016) | $53.1 \pm 0.2$ | $18.3 \pm 0.1$ | $44.2 \pm 0.7$ | $11.8 \pm 0.1$ | $55.5 \pm 0.4$ | $46.8 \pm 0.6$ | 38.3 |
| CDANN (Li et al., 2018b) | $54.6 \pm 0.4$ | $17.3 \pm 0.1$ | $43.7 \pm 0.9$ | $12.1 \pm 0.7$ | $56.2 \pm 0.4$ | $45.9 \pm 0.5$ | 38.3 |
| **Ours** (SRA) | $64.2 \pm 0.3$ | $21.6 \pm 0.9$ | $50.8 \pm 1.1$ | $13.3 \pm 0.8$ | $64.4 \pm 0.1$ | $53.0 \pm 0.4$ | 44.5 |
| SWAD Cha et al. (2021) | $66.0 \pm 0.1$ | $22.4 \pm 0.3$ | $53.5 \pm 0.1$ | $16.1 \pm 0.2$ | $65.8 \pm 0.4$ | $55.5 \pm 0.3$ | 46.5 |
| SWAD + DANN | $64.3 \pm 0.1$ | $21.9 \pm 0.6$ | $52.6 \pm 0.2$ | $15.5 \pm 0.2$ | $65.3 \pm 0.1$ | $54.5 \pm 0.1$ | 45.7 |
| SWAD + CDANN | $64.3 \pm 0.2$ | $21.9 \pm 0.4$ | $52.5 \pm 0.0$ | $15.6 \pm 0.0$ | $65.3 \pm 0.1$ | $54.4 \pm 0.2$ | 45.7 |
| **Ours** (SRA + SWAD) | $67.4 \pm 0.1$ | $23.5 \pm 0.2$ | $55.0 \pm 0.1$ | $15.9 \pm 0.2$ | $67.2 \pm 0.2$ | $56.6 \pm 0.1$ | 47.6 |
| **Ours** (SRA + SWAD + Ensemble) | $\mathbf{68.7} \pm 0.0$ | $\mathbf{24.0} \pm 0.2$ | $\mathbf{56.3} \pm 0.0$ | $\mathbf{16.7} \pm 0.0$ | $\mathbf{68.5} \pm 0.0$ | $\mathbf{57.8} \pm 0.0$ | **48.7** |

