# OpenReview forum: "Understanding Domain Generalization: A View of Necessity and Sufficiency"
_ICLR.cc/2025/Conference — Submitted to ICLR 2025_

### Official Review · Reviewer_RzUW · 2024-11-02

**Soundness:** 2
**Presentation:** 2
**Contribution:** 2
**Rating:** 5
**Confidence:** 2

**Summary:**

This paper first studies the conditions for the existence and learning of an optimal hypothesis. The authors analyze the existing methods on whether they meet the necessary conditions 3.3 and 3.7. They further incorporate the sufficient representation constraint via ensemble learning and present a novel representation alignment strategy that can enforce the necessary conditions. They verify the effectness of the proposed method on five datasets against four baseline methods.

**Strengths:**

The paper is well-written and mathematical sound.  The proposed necessary conditions can be used to verify whether a method is good or not.

**Weaknesses:**

The experimental results lack too much baseline to compare with, e.g., IRM, VREx, IIB, IB-IRM mentioned in section 4.2.

**Questions:**

There is a line of works that also use mutual information for invariant representation. The authors have not mentioned and compared with them.

Cha, J., Lee, K., Park, S., & Chun, S. (2022). Domain Generalization by Mutual-Information Regularization with Pre-trained Models. ArXiv, abs/2203.10789.

---

> ### Author Response · Authors · 2024-11-23
>
> We sincerely appreciate your thorough and insightful feedback. Below, we outline our efforts to address the valuable points you have raised.
>
> &nbsp;
>
> **Weakness:** The experimental results lack too much baseline to compare with, e.g., IRM, VREx, IIB, IB-IRM mentioned in section 4.2.
>
> *Reply:*
>
> We have included a baseline consisting solely of SRA without SWAD. While SRA achieves the best average performance on the benchmark without SWAD, it does not consistently outperform across all datasets compared to the settings with SWAD.
>
> Additionally, we have included further experimental results in Section C.4, comparing our method with other baselines from DomainBed.
>
> For the baselines with SWAD, due to time constraints, we have not yet completed the experiments on the DomainNet dataset. We will update the results as soon as they are available.
>
> &nbsp;
>
> **Why we selected DANN and CDANN as the main baselines**
>
> **First**: DANN and CDANN are the two most closely related methods, as they represent specific cases of our approach. Specifically, SRA (ours), DANN, and CDANN all utilize $\mathcal{H}$-divergence for alignment.
>
> The key difference lies in what they align:
> - DANN aligns the entire domain representation.
> - CDANN aligns class-conditional representations.
> - SRA align subspace-conditional representations (e.g., $\left| \mathcal{M}\right|$ subspaces per class).
>
> CDANN is equivalent to SRA with $\left| \mathcal{M}\right| = 1$.
>
>
> **Second:** One of our main messages is the recommendation to design domain generalization (DG) algorithms that strive for the sufficient conditions but making sure that the necessary conditions are satisfied.
>
> As presented in Theorems 4.2 and 5.1, DANN and CDANN suffer from a **trade-off** between optimizing representation-alignment (striving for the sufficient condition 3.5) and minimizing training domain losses (necessary condition 3.3), while SRA addresses this issue effectively.
>
> Note that if Necessary Condition 3.3 is violated, then by Theorem 3.8, Necessary Condition 3.7 is also violated.
>
>
> At a higher level, DANN and CDANN aim to achieve invariant representations (a sufficient condition) but impose problematic constraints, as the trade-off with training domain losses can harm the necessary conditions for generalization. In contrast, SRA imposes well-structured constraints, allowing it to simultaneously encourage alignment and minimize training domain losses.
> This validates our theoretical conclusions: enforcing good sufficient conditions (as in SRA) while promoting necessary conditions can significantly improve generalization.
>
> &nbsp;
>
> **Questions** There is a line of works that also use mutual information for invariant representation. The authors have not mentioned and compared with them.
>
> Cha et al. (2022) focus on maximizing the mutual information between the pre-trained model and the trained model, which differs from our objective, as we aim to maximize the information between the learned representation and the causal factors.

---

> > ### Comment · Reviewer_RzUW · 2024-12-01
> >
> > Thank the authors for the response. I have carefully read the rebuttal. I understand the claim of the authors. I will keep the score.

---

### Official Review · Reviewer_tHNP · 2024-11-03

**Soundness:** 2
**Presentation:** 1
**Contribution:** 2
**Rating:** 3
**Confidence:** 4

**Summary:**

This paper studies the sufficient and necessary conditions for the success of domain generalization (DG) problems. The authors demonstrate that under the assumption of a covariate shift setting, where causal support is covered by the training domains, having a sufficient number of diverse training domains or learning an invariant representation function will enable a DG algorithm to generalize effectively. However, as meeting these sufficient conditions is often impractical, the authors introduce necessary conditions: specifically, that the optimal hypothesis for the training domains must also be optimal for the global domain, and that the representation function must serve as a sufficient representation. Furthermore, the authors apply these conditions to analyze existing DG approaches. They also propose an algorithm designed to learn sufficient invariant representations by partitioning each training domain into subdomains based on their labels.

**Strengths:**

1. The use of sufficient and necessary conditions to identify potential failures in previous DG works is both interesting and significant.

2. Partitioning each training domain according to its labels and aligning the corresponding partitions across different training domains seems to be an effective strategy.

**Weaknesses:**

1. I have concerns regarding the causal relationships between variables in this paper, particularly with verifying that the label $Y$ is conditionally independent of $E$ given $Z\_c$. This seems essential to justify the covariate shift assumption made in Assumption 3.1. Another potentially related concern is whether your analysis is constrained solely to covariate shift. Under covariate shift, $P^{e}(X)\neq P^{e'}(X)$ for different domains, while $P^{e}(Y|X=x)= P^{e'}(Y|X=x)$ holds for any $e,e'\in\mathcal{E}$. However, what if conditional shift occurs, namely $P^{e}(Y|X=x)\neq P^{e'}(Y|X=x)$? Conditional shift is also common in DG and could imply that $Y$ may still depend on $E$ even when conditioned on $Z\_c$?

2. The theoretical development and writing in this paper lack rigor, largely due to many undefined or inconsistent notations. Below are some examples, though there are additional instances throughout the paper: 1) In Eq.(1) $\mathcal{F}^*$ is defined using the undefined term $\mathcal{L}$. I suggest defining the loss before introducing $\mathcal{F}^*$. 2) The notation $\mathcal{F}_{\mathbb{P}^{e_i}}$ first appears in Line 204 without definition; it is only defined later in Line 285.  3) In Line 227, it is unclear what is meant by $\mathbb{P}(g(x))=\mathbb{P}(Y|z_c)$. Since $g:\mathcal{X}\to\mathcal{Z}$, how can the probability of the representation output be equal to the conditional probability of the label? 4) In Lines 227, 232, 240, and 243, notations such as $g\in\mathcal{G}\_c$, $g\_c$ within the set $\mathcal{G}\_c$, $g_c\in\mathbf{G}\_c$ and $g\in\mathcal{G}\_c$ are used interchangeably. Are they intended to mean the same thing? 5) In Definition 3.7, the function set $\Phi$ is not defined. I assume $\mathcal{G}\_c$ refers to a function set mapping $\mathcal{X}$ to $\mathcal{Z}$, so $\phi\circ g:\mathcal{X}\to\mathcal{Z}$. In that case, what are the domain and range of $g\in\mathcal{G}\_s$? 6) In Line 252, $\mathcal{L}(h\circ g(x),y)$ is used, where, where the input space of $\mathcal{L}$ is $\Delta\_{|\mathcal{Y}|}\times\mathcal{Y}$. This notation contradicts other uses in the paper, such as $\mathcal{L}(f,\mathbb{P}^e)$. 7) In Line 452, $\Gamma:\mathcal{X}\to\mathcal{M}$ is defined, and then $\Gamma=f$ is set in Line 485. Given that $f:\mathcal{X}\to\Delta\_{|\mathcal{Y}|}$, does this imply $\mathcal{M}=\Delta\_{|\mathcal{Y}|}$? This seems inconsistent, as $\mathcal{M}$ was previously described as a set of indices. 8) In Line 486, what is the intended meaning of $m(i)$? 9) In Line 828, the expression $h(Y|g\_c(x))$ is unclear. Since $h:\mathcal{Z}\to\Delta\_{|\mathcal{Y}|}$, the equation $h(Y|g\_c(x))=\mathbb{P}(Y|z\_c)$ is confusing. 10) In Eqs. (9–12) in the Appendix, there seems to be an inconsistency. In Eq. (9), a conditional expectation is taken with respect to  $X=x$. How, then, does $\mathbb{P}^{e'}(X=x|z_c,z_e)$ appear in Eq. (10)? Did you mean to write $\mathbb{P}^{e'}(Z_c=z_c,Z_e=z_e|X=x)$ here? 11) In Line 899, the sentence "we construct a new function $h\_\phi=h\circ\phi$, where $h\circ g\_c$ belongs to $\mathcal{F}\_{\mathbb{P}^e,g\_c}$." is unclear. Could you clarify this?

There are additional such issues throughout the paper; I recommend a thorough proofread to fix these inconsistencies.

3. The lack of rigor in the writing raises concerns about the theoretical development in this paper. For example, in the proof of Theorem 3.6: 1) A minor note, Theorem A.5 restates Theorem 3.8, not Theorem 3.6 as indicated; 2) You let $x=\psi(z_c,z_e)$,  in the proof, which contradicts the setup in Line 116 and Proposition A.3, where $x=\psi(z_c,z_e,u_x)$. Why is $u_x$ omitted here? 3) More importantly, Eqs. (9–12) are confusing. While $f_c$ is defined as the risk minimizer under $\mathbb{P}^e$, it is unclear why it would also minimize the risk of every data from $\mathbb{P}^{e'}$; 4) By the way, the loss notation $\ell(f(X),\mathbb{P}(Y|z_c))$ used in Eqs. (9–12) differs from the notation $\ell(f(X),Y)$ used in the main paper, and as noted previously, the conditional expectation in this proof is unclear, as it seems to be taken with $X=x$ fixed. Could you clarify the exact conditional distribution over which this expectation is calculated? Please provide further clarification on the proof of Theorem 3.6.

4. Related to the previous concern, in Theorem 5.1, point (i), it seems that the intention is to provide an upper bound for the target generalization loss, but what is actually bounded is the average risk over the training domains. Additionally, in the proof of Theorem 5.1, could you clarify the steps between Eq.(19) and (20)? Specifically, it is unclear to me why the LHS in Eq. (20) includes a factor of $\frac{1}{|\mathcal{E}_{tr}|}$ while the RHS does not.

Overall, the paper is very unclear in its current state, and some theoretical results seem quite trivial (e.g., Theorem 3.4, Corollary 4.1). A significant revision is needed to improve clarity and rigor.


Minor comments:

1. Note that the $\mathcal{H}$-divergence was originally proposed in Ben-David et al (2006), not by Zhao et al.(2019).

Shai Ben-David, et al. "Analysis of representations for domain adaptation." Advances in neural information processing systems 19 (2006).

2. In Line 051, the sentence seems incomplete, as there is an extra "which."

3. In Corollary 4.1, $0\geq \delta\geq 1\Longrightarrow 0\leq \delta\leq 1$.

4. In Lines 453–455, the sentence "Let $\mathbb{P}^e_m$ be ..." defines $\mathbb{P}^e_m$ twice. Please rewrite this for clarity.

**Questions:**

What is the exact difference between an invariant representation function and a sufficient representation function? This question may stem from the undefined function $\phi$; could you clarify its exact role?

---

> ### Author Response · Authors · 2024-11-24
> **Responses for Weakness 1**
>
> We sincerely appreciate your thorough and insightful feedback. Below, we outline our efforts to address the valuable points you have raised.
>
> &nbsp;
>
> **Weakness 1**: Conditional shift
>
> We agree with the reviewer that conditional shift is also common in domain generalization (DG). However, our current setting is restricted to covariate shift. It is worth noting that our setting is more practical, as our analysis focuses on ***limited and finite domains***. Within this context, even when focusing solely on covariate shift, the DG problem remains far from being fully addressed.

---

> ### Author Response · Authors · 2024-11-24
> **Responses for Weakness 2 (1/2)**
>
> &nbsp;
>
> **Weakness 2**:
>
> &nbsp;
>
> **W1** *$\mathcal{F}^\*$ is defined using the undefined term $\mathcal{L}$. I suggest defining the loss before introducing $\mathcal{F}^\*$ .*
>
> *Reply:*
>
>  Following the reviewer's suggestion, we have revised the manuscript and moved Equation (2) before Equation (1) for better readability.
>
> Additionally, to enhance clarity, we rigorously define the loss function:
>   $\ell:\mathcal{Y}\_{\Delta} \times \mathcal{Y} \rightarrow \mathbb{R}$
>   where $\ell\left(f(x), y\right)$ with $f(x) \in \mathcal{Y}_{\Delta}$ and $y \in \mathcal{Y}$ specifies the loss (e.g., cross-entropy) for assigning a data sample $x$ to the class $y$ using the hypothesis $f$;
>
> and updated all related discussions and proofs to reflect the new definition of $\ell$.
>
> &nbsp;
>
> **W2** *The notation$\mathcal{F}_{\mathbb{P}^{e_i}}$ first appears in Line 204 without definition; it is only defined later in Line 285.*
>
> *Reply:*
>
> In the revised version, we introduce $\mathcal{F}_{\mathbb{P}^{e_i}}$  in Definition 3.3 before using it.
>
> &nbsp;
>
> **W3** *In Line 227, it is unclear what is meant by $\mathbb{P}(g(x)) = \mathbb{P}(Y\mid z_c)$. Since $g: \mathcal{X}\rightarrow \mathcal{Z}$, how can the probability of the representation output be equal to the conditional probability of the label?*
>
> *Reply:*
>
> We apologize for the error. It should be $\mathbb{P}(Y\mid g(x)) = \mathbb{P}(Y\mid z_c)$. This has been corrected in the revised version.
>
> &nbsp;
>
> **W4** *In Lines 227, 232, 240, and 243, notations such as $g\in\mathcal{G}_c$, $g_c$ within the set $\mathcal{G}_c$, $g_c\in G_c$,  and $g\in \mathcal{G}_c$ are used interchangeably. Are they intended to mean the same thing?*
>
> *Reply:*
>
> All of these terms are intended to convey the same meaning, i.e., $g \in \mathcal{G}_c$. We have clarified this in the revised version.
>
> &nbsp;
>
> **W5** *In Definition 3.7, the function set $\Phi$ is not defined. I assume $\mathcal{G}_c$ refers to a function set mapping $\mathcal{X}$ to $\mathcal{Z}$, so $\phi \circ g: \mathcal{X} \rightarrow\mathcal{Z}$. In that case, what are the domain and range of $g\in \mathcal{G}_s$*
>
> *Reply:*
>
> All $\mathcal{G}, \mathcal{G}_c, \mathcal{G}_s$ refer to sets of functions mapping $\mathcal{X}$ to $\mathcal{Z}$. The relationship between them is as follows: $\mathcal{G}_c \subseteq \mathcal{G}_s \subseteq \mathcal{G}$.
>
> $\phi$ is a function that maps $\mathcal{Z}$ to $\mathcal{Z}$.
>
> We have updated the definitions of $\mathcal{G}_c$, $\mathcal{G}_s$, and $\phi$ in the revised version for clarity.
>
> &nbsp;
>
> **W6** *In Line 252, $\mathcal{L}(h\circ g(x),y)$ is used, where, where the input space of $\mathcal{L}$
>  is $\Delta_{\left| \mathcal{Y}\right|}\times\mathcal{Y}$. This notation contradicts other uses in the paper, such as $\mathcal{L}(f,\mathbb{P}^e)$.*
>
> *Reply:*
>
> We have updated $\mathcal{L}(h\circ g(x),y)$ on line 252 to $\mathcal{L}(f,\mathbb{P}^e)$. The manuscript, including both the main text and the appendix, has been revised accordingly based on this definition:
>
> $\mathcal{L}\left(f,\mathbb{P}^e\right):=\mathbb{E}_{\left(x,y\right)\sim\mathbb{P}^e}\left[\ell\left(f\left(x\right),y\right)\right]$
>
> &nbsp;
>
> **W7** *In Line 452,  is defined, and then $\Gamma=f$ is set in Line 485. Given that $f: \mathcal{X}\rightarrow \Delta_{\left| \mathcal{Y}\right|}$, does this imply $\mathcal{M}=\Delta_{\left| \mathcal{Y}\right|}$? This seems inconsistent, as was previously described as a set of indices.*
>
> *Reply:*
>
> We apologize for the unclear description of $\mathcal{M}$. $\mathcal{M}$ is initially introduced as a general set of subspace indices, where the values of the subspace indices depend on the design of the projector $\Gamma: \mathcal{X} \to \mathcal{M}$.
>
> We have revised the description of \(\mathcal{M}\) and \(\Gamma\) on line 452 for clarity as follows:
>
> "We can define a projector $\Gamma = f$, which induces a set of subspace indices $\mathcal{M}=\\{m=\hat{y}\mid \hat{y}=f(x), x\in\bigcup\_{e\in\mathcal{E}\_{tr}}\text{supp}\mathbb{P}^{e} \\}\subseteq \Delta_{\left | \mathcal{Y}\right |}$. As a result, given subspace index $m\in\mathcal{M}$, $\forall i \in \mathcal{Y}, \mathbb{P}^{e}\_{\mathcal{Y},m}(Y=i) = \mathbb{P}^{e\'}\_{\mathcal{Y},m}(Y=i) = \sum_{x \in f^{-1}(m)}\mathbb{P}(Y=i\mid x) = m[i]$. "
>
> &nbsp;
>
> **W8** *In Line 486, what is the intended meaning of $m(i)$?*
>
> *Reply:*
>
> Since $m \in \Delta_{\left| \mathcal{Y} \right|}$, $m[i]$ represents the $i$-th element of the vector $m$. For clarity, we have changed the notation from $m(i)$ to $m[i]$.
>
>
>
>
> &nbsp;
>
> **W9** *In Line 828, the expression $h(Y\mid g(x))$ is unclear. Since $h: \mathcal{Z}\rightarrow \Delta_{\left| \mathcal{Y}\right|}$, the equation $h(Y\mid g(x))=\mathcal{P}(Y\mid z_c)$ is confusing.*
>
> *Reply:*
>
> We apologize for the error. It should be $h(g(x)) = \mathbb{P}(Y\mid z_c)$. This has been corrected in the revised version.
>
> &nbsp;

---

> ### Author Response · Authors · 2024-11-24
> **Responses for Weakness 2 (2/2)**
>
> **W10** *In Eqs. (9–12) in the Appendix, there seems to be an inconsistency. In Eq. (9), a conditional expectation is taken with respect to $X=x$. How, then, does $\mathbb{P}^{e\'}(X=x\mid z\_c,z\_e)$ appear in Eq. (10)? Did you mean to write $\mathbb{P}^{e'}(Z_c=z_c,Z_e=z_e\mid X=x)$ here?*
>
> *Reply:*
>
> Based on structural causal model (SCM) depicted in Figure.1 we have a distribution (domain) over the observed variables $(X,Y)$ given the environment $E=e \in \mathcal{E}$:
>
> $$
> \mathbb{P}^e(X,Y)= \int\_{\mathcal{Z}\_c}\int\_{\mathcal{Z}\_e}\mathbb{P}^{e}(z\_c)\mathbb{P}^{e}(z\_e)\int\_{\mathcal{X}}\mathbb{P}^{e}(X=x\mid z\_c, z\_e)\int\_{\mathcal{Y}}\mathbb{P}^{e}(Y=y\mid z\_c) d\_{z\_c} d\_{z\_e} d\_x d\_y
> $$
>
> We have revised the proof of Theorem A.6 (Theorem 3.6 in the main paper) and provided a detailed derivation from lines 980 to 985.
>
> &nbsp;
>
> **W11** *In Line 899, the sentence "we construct a new function $h\_\phi=h \circ \phi$ where $h \circ g\_c$ belong to $\mathcal{F}\_{\mathbb{P}\_e},g\_c$." is unclear. Could you clarify this?*
>
> *Reply:*
>
> We have revised the proof of the "if" direction in Theorem A.7 (Theorem 3.8 in the main paper) for clarity as follows:
>
> If $g\in \mathcal{G}\_s$, we have:
>
> - (1)There exists a function $\phi$ such that $\phi \circ g \in \mathcal{G}\_c$, which implies the existence of a $g\_c\in \mathcal{G}\_c$ such that $\phi \circ g = g\_c$.
>
> - (2) By the definition of $g\_c \in \mathcal{G}\_c$, we can always find a classifier $h$ such that $h \circ g\_c \in \mathcal{F}^\*$.
>
> Recall that the definition of $\mathcal{F}^{B}\_{\mathbb{P}^e, g}$ implies that, given $g$, we perform an oracle search for a classifier $h' \in \mathcal{H}\_{\mathbb{P}^e, g}$ such that $h' \circ g$ achieves the best generalization across any domain $e' \in \mathcal{E}$.
>
>
> From (1) and (2) we have $h\circ g\_c = h\circ \phi \circ g \in \mathcal{F}^\*$. Therefore, **we can construct classifier $h\_{\phi} = h \circ \phi$, then $h\_{\phi} \circ g =  h\circ \phi \circ g =h\circ g\_c \in \mathcal{F}^\*$.** (which implies $h \circ g\_c $ also belongs to $ \mathcal{F}\_{\mathbb{P}\_e},g\_c$)
>
> Since $h\_{\phi} \circ g \in \mathcal{F}^\*$ is the optimal hypothesis across all domains, it is also the optimal hypothesis for the specific domain $\mathbb{P}^e$, i.e., $h\_{\phi} \circ g \in \mathcal{F}\_{\mathbb{P}^e, g}$. This implies that $h\_{\phi} \in \mathcal{H}\_{\mathbb{P}^e, g}$. Consequently, the set $\mathcal{F}^{B}\_{\mathbb{P}^e, g} \subseteq \mathcal{F}^\*$.

---

> ### Author Response · Authors · 2024-11-24
> **Responses for Weakness 3**
>
> **Weakness 3: The lack of rigor in the writing raises concerns about the theoretical development in this paper.**
>
> We apologize for the mistake and the lack of clarity in the notation and proof. We have revised the manuscript as follows:
>
> &nbsp;
>
>  ***Theorem A.5 restates Theorem 3.8***: We have updated Theorem A.5 to Theorem 3.6.
>
> &nbsp;
>
> ***Why is $u_x$ omitted here***:  We have included $u_x$ in all the proofs for clarity.
>
> &nbsp;
>
> ***Eqs. (9–12) are confusing***: We have revised the proof and provided a detailed derivation of these equations in the revised version, spanning lines 980 to 1036  (highlighted in blue). We apologize for the lengthiness of the explanation.
>
> &nbsp;
>
> ***Why $f_c$ would also minimize the risk of every data from $\mathcal{P}^{e\'}$?***
>
> We have revised the proof and provided a detailed explanation for the generalization of $f_c$ in the revised version, spanning lines 1036 to 1054 (highlighted in blue).
>
> &nbsp;
>
> ***The loss notation***: we redefine the loss function:
>   $\ell:\mathcal{Y}\_{\Delta} \times \mathcal{Y} \rightarrow \mathbb{R}$
>   where $\ell\left(f(x), y\right)$ with $f(x) \in \mathcal{Y}_{\Delta}$ and $y \in \mathcal{Y}$ specifies the loss (e.g., cross-entropy) for assigning a data sample $x$ to the class $y$ using the hypothesis $f$;
>
> and updated all related discussions and proofs to reflect the new definition of $\ell$.
>
> &nbsp;
>
> ***Finally,*** in the previous version, we implicitly used some properties of $\mathcal{G}_c$, which caused ambiguity in the proof. To address this, we have added Corollary A.5 (Invariant Representation Function Properties) and referenced it when proving Theorem 3.6 for clarity.

---

> ### Author Response · Authors · 2024-11-24
> **Responses for Weakness 4**
>
> **Weakness 4: There are several concerns regarding Theorem 5.1**
>
> &nbsp;
>
>
> **W1:** Related to the previous concern, in Theorem 5.1, point (i), it seems that the intention is to provide an upper bound for the target generalization loss, but what is actually bounded is the average risk over the training domains
>
> *Reply:*
>
> The purpose of Theorem 5.1 (i) is not to provide an upper bound for the target generalization loss, as both the LHS and RHS pertain to the training domains.
>
> Our goal is to design an approach that strives  for the sufficient conditions (specifically, learning *invariant representation function* through a representation-alignment strategy) while ensuring that the necessary conditions (Conditions 3.3 and 3.7) are also met.
>
> &nbsp;
>
> We would like to recall the theoretical result presented in Theorem 4.2:
>
> $
> D\left(\mathbb{P}\_{\mathcal{Y}}^{e},\mathbb{P}\_{\mathcal{Y}}^{e'}\right) \leq D\left ( g\_{\\#}\mathbb{P}^{e},g\_{\\#}\mathbb{P}^{e'} \right )+\mathcal{L}\left ( f,\mathbb{P}^{e} \right )+\mathcal{L}\left ( f,\mathbb{P}^{e'} \right )
> $
>
> This result suggests that if there is a substantial discrepancy $D\left(\mathbb{P}\_{\mathcal{Y}}^{e},\mathbb{P}\_\{\mathcal{Y}}^{e'}\right)$ in the label marginal distribution across training domains, enforcing *representation alignment* by minimizing $D\left( g\_{\\#}\mathbb{P}^{e}, g\_{\\#}\mathbb{P}^{e'} \right)$ will increase *domain losses* $\mathcal{L}\left( f, \mathbb{P}^{e} \right) + \mathcal{L}\left( f, \mathbb{P}^{e'} \right).$. (as the LHS > 0, the RHS cannot be optimized to zero)
>
> In Theorem 5.1 (ii), we demonstrate that, with appropriately organized subspaces, it is possible to simultaneously optimize both "subspace representation alignment" and "subspace training domain loss" without any trade-offs.
>
> However, one of our necessary conditions is to achieve the optimal hypothesis on the original training domains (condition 3.3).
>
> Referring back to Theorem 5.1 (i), the LHS represents the total general loss over the original training domains, which is upper-bounded by the RHS—the sum of the subspace training loss and the subspace alignment loss. Theorem 5.1 (ii) shows that the RHS of Theorem 5.1 (i) can be optimized to zero, implying that the LHS also becomes zero. This results in achieving the optimal hypothesis for the training domains.
>
> &nbsp;
>
> **W2.** Additionally, in the proof of Theorem 5.1, could you clarify the steps between Eq.(19) and (20)? Specifically, it is unclear to me why the LHS in Eq. (20) includes a factor of $\frac{1}{\left |\mathcal{E}_{r} \right |}$ while the RHS does not.
>
> *Reply:*
>
> It should be $\left | \mathcal{E}\_{r}\right |$ instead of $\frac{1}{\left| \mathcal{E}\_{r} \right |}$ on the left-hand side (LHS). We apologize for this error. The manuscript has been updated in both the main text and the appendix, and the proof (steps between Eq.(19) and (20)) of this theorem (A11) is now provided in greater detail (highlighted in blue).
>
> &nbsp;

---

> ### Author Response · Authors · 2024-11-24
> **Responses for Question**
>
> **Question:** *What is the exact difference between an invariant representation function and a sufficient representation function? This question may stem from the undefined function $\phi$; could you clarify its exact role?*
>
> &nbsp;
>
> The family of invariant representation functions $\mathcal{G}_c$, is set of flexible functions, which map a sample $x = \psi(z_c, z_e, u_x)$ to a set of ***equivalent*** causal factors $\{z'_c \in \mathcal{Z}_c \mid \mathbb{P}(Y \mid z_c) = \mathbb{P}(Y \mid z'_c)\}$, rather than requiring an exact mapping to $z_c$.  (Further details are provided in Corollary A.5 (Invariant Representation Function Properties); and the relationship between them is as follows: $\mathcal{G}_c \subseteq \mathcal{G}_s \subseteq \mathcal{G}_c$)
>
> &nbsp;
>
> However, achieving $g_c\in \mathcal{G}_c$ is often infeasible in practice, because it requires the knowledge of all domains. We therefore shift the attention to studying a new class of representation function i.e., sufficient representation functions $\mathcal{G}_s$.
>
> The definition of $\mathcal{G}_s$ states that there exists a function $\phi: \mathcal{Z} \rightarrow \mathcal{Z}$ such that $(\phi \circ g) \in \mathcal{G}_c$.
>
> This means that for $g \in \mathcal{G}_s$, $g(x)$ retains all information about the (equivalent) causal features of $x$, as it is possible to apply the mapping $\phi: \mathcal{Z} \rightarrow \mathcal{Z}$ to recover the invariant correlation from $g$.
>
> &nbsp;
>
> Note that an identity function also satisfies the sufficient representation function condition. However, this is not typically the case for representation learning, as they generally map high-dimensional inputs to lower-dimensional spaces, inherently reducing the amount of information. Furthermore, when applying DG strategies, the process often further discards information to extract abstract (or ideally causal) features from the input.
>
>
> &nbsp;
>
> **Why is awareness of the sufficient representation function important?**
>
> Theorem 3.8 demonstrates that even with complete knowledge of all domains, if $g \notin \mathcal{G}_s$, it is impossible to find a classifier $h$ such that $h \circ g \in \mathcal{F}^\*$. This is because $g$ fails to capture the (equivalent) causal factors and instead relies on spurious features, which are unstable as they vary across domains.
>
>
> The DG literature often overlooks the importance of sufficient representation function (Condition 3.7) in real-world scenarios. This oversight occurs because algorithms that satisfy sufficient conditions are assumed to automatically fulfill the sufficient representation function (Condition 3.7). However, achieving sufficient conditions is almost impossible in finite-domain settings. DG analysis, which often focuses on idealized settings where sufficient conditions are more likely to be satisfied, can lead to misleading assumptions about Condition 3.7 in practical applications. For instance, methods like IIB (Li et al., 2022a) and IB-IRM (Ahuja et al., 2021), which incorporate the information bottleneck principle, may inadvertently harm the sufficient representation function (Condition 3.7).

---

> ### Author Response · Authors · 2024-11-28
> **Gentle Reminder: Further Discussion**
>
> Dear reviewer tHNP,
>
> We really appreciate your acknowledgment of our contribution, which makes us feel it is our obligation to improve the quality of this paper. We are looking forward to hearing from your further opinions about our rebuttal. We cherish the opportunity to polish our paper through this interactive discussion and hope you can spare a little time to take a look.
>
> We would be glad to have further discussions if there are any suggestions or questions. Thank you again for your effort and valuable comments.
>
> Best regards,
>
> The Authors.

---

> > ### Comment · Reviewer_tHNP · 2024-12-03
> >
> > I would like to thank the authors for their detailed responses and sincerely apologize for my late engagement during the discussion phase. However, I still think this paper requires significant revisions and another round of review, so I intend to keep my score at this point.

---

### Official Review · Reviewer_ZUgG · 2024-11-04

**Soundness:** 1
**Presentation:** 2
**Contribution:** 2
**Rating:** 3
**Confidence:** 4

**Summary:**

Authors analyze the sufficient conditions and necessary conditions of achieving domain generalization. They further propose an algorithm to learn sufficient invariant representation. Experiments show their effectiveness.

**Strengths:**

The systematic analyses of sufficient and necessary conditions are novel.

**Weaknesses:**

1. Although the sufficient representation condition is a necessary condition, it is easy to satisfy using an identical function. The incorporation of this constraint is meaningful and beneficial only if there is empirical evidence that most of current algorithms are far from satisfying this.
2. As a core part of the paper, sufficient representation constraint is only implemented by ensemble learning. The connection between them, although explained in the last paragraph of Section 5.1, seems farfetched.
3. Only several basic methods are compared with the proposed method. You may refer to the github repo of DomainBed for more baselines.

Minor issues:

- In Line 139, "the the".
- In Line 145, "while vary"
- In Figure 2, the part of $\mathcal{F}\_{P^{e_2}}\cap\mathcal{F}\_{P^{e_3}}-\mathcal{F}\_{P^{e_1}}$ seems absent in the Venn diagram.

##

**Questions:**

Please refer to weaknesses.

---

> ### Author Response · Authors · 2024-11-23
> **Domainbed Benchmark**
>
> We sincerely appreciate your thorough and insightful feedback. Below, we outline our efforts to address the valuable points you have raised.
>
> &nbsp;
>
> **Weakness:** Only several basic methods are compared with the proposed method. You may refer to the github repo of DomainBed for more baselines.
>
> *Reply:*
>
> We have included a baseline consisting solely of SRA without SWAD. While SRA achieves the best average performance on the benchmark without SWAD, it does not consistently outperform across all datasets compared to the settings with SWAD.
>
> Additionally, we have included further experimental results in Section C.4, comparing our method with other baselines from DomainBed.
>
> For the baselines with SWAD, due to time constraints, we have not yet completed the experiments on the DomainNet dataset. We will update the results as soon as they are available.
>
> &nbsp;
>
> ***Why we selected DANN and CDANN as the main baselines***
>
> ***First***: DANN and CDANN are the two most closely related methods, as they represent specific cases of our approach. Specifically, SRA (ours), DANN, and CDANN all utilize $\mathcal{H}$-divergence for alignment.
>
> The key difference lies in what they align:
> - DANN aligns the entire domain representation.
> - CDANN aligns class-conditional representations.
> - SRA align subspace-conditional representations (e.g., $\left| \mathcal{M}\right|$ subspaces per class).
>
> CDANN is equivalent to SRA with $\left| \mathcal{M}\right| = 1$.
>
>
> ***Second:*** One of our main messages is the recommendation to design domain generalization (DG) algorithms that strive for the sufficient conditions but making sure that the necessary conditions are satisfied.
>
> As presented in Theorems 4.2 and 5.1, DANN and CDANN (two representative methods of the representation alignment strategy) suffer from a **trade-off** between optimizing representation-alignment (striving for the sufficient condition 3.5) and minimizing training domain losses (sufficient condition 3.3), while SRA addresses this issue effectively. Note that if Necessary Condition 3.3 is violated, then by Theorem 3.8, Necessary Condition 3.7 is also violated.
>
> At a higher level, DANN and CDANN aim to achieve invariant representations (a sufficient condition) but impose problematic constraints, as the trade-off with training domain losses can harm the necessary conditions for generalization. In contrast, SRA imposes well-structured constraints, allowing it to simultaneously encourage alignment and minimize training domain losses.
> This validates our theoretical conclusions: enforcing good sufficient conditions (as in SRA) while promoting necessary conditions can significantly improve generalization.

---

> ### Author Response · Authors · 2024-11-24
> **Responses for Weakness 1 and Weakness 2**
>
> &nbsp;
>
> **Weakness 1:** Although the sufficient representation condition is a necessary condition, it is easy to satisfy using an identical function. The incorporation of this constraint is meaningful and beneficial only if there is empirical evidence that most of current algorithms are far from satisfying this.
>
> *Reply:*
>
> We respectfully disagree with the reviewer on this point. While an identity function also satisfies the sufficient representation function condition.  this is not typically the case for representation learning, as they generally map high-dimensional inputs to lower-dimensional spaces, inherently reducing the amount of information. Furthermore, when applying DG strategies, the process often further discards information to extract abstract (or ideally causal) features from the input.
>
> &nbsp;
>
> ***Theoretical and Empirical Evidence for the Significance and Benefits of the Sufficient Representation Function Condition***
>
> &nbsp;
>
> ***From a theoretical perspective:*** As demonstrated in Theorems 4.2, we show that a family of DG algorithms, specifically those employing representation alignment strategies, suffer from a **trade-off** between optimizing representation alignment (striving for the sufficient condition 3.5) and minimizing training domain losses (necessary condition 3.3).
>
> Note that if Necessary Condition 3.3 is violated, then by Theorem 3.8, Necessary Condition 3.7 is also violated.
>
> ***From Empirical perspective:***  In Theorem 5.1, we demonstrate that, with appropriately organized subspaces, it is possible to simultaneously optimize both representation alignment and training domain loss (optimal hypothesis for training domains condition) without any trade-offs.
>
> On average, SRA outperforms DANN/CDANN by approximately 2% without SWAD and 1.8% with ensemble learning, representing a significant improvement in the DG benchmark. These results highlight that effectively ensuring the necessary conditions (3.3 and 3.7) can substantially enhance generalization.
>
> Furthermore, as demonstrated in the additional results in Section C.4, the DG baselines—which are theoretically designed to improve generalization—fail to consistently outperform the simple ERM baseline across all settings. This provides empirical evidence that most current algorithms fall short of satisfying these conditions.
>
> ***Additional discussion:*** The DG literature often overlooks the importance of sufficient representation function (Condition 3.7) in real-world scenarios. This oversight occurs because algorithms that satisfy sufficient conditions are assumed to automatically fulfill the sufficient representation function (Condition 3.7). However, achieving sufficient conditions is almost impossible in finite-domain settings. DG analysis, which often focuses on idealized settings where sufficient conditions are more likely to be satisfied, can lead to **misleading** assumptions about Condition 3.7 in practical applications. For instance, methods like IIB (Li et al., 2022a) and IB-IRM (Ahuja et al., 2021), which incorporate the information bottleneck principle, may inadvertently harm the sufficient representation function (Condition 3.7).
>
> &nbsp;
>
> **Weakness 2:** As a core part of the paper, sufficient representation constraint is only implemented by ensemble learning. The connection between them, although explained in the last paragraph of Section 5.1, seems farfetched.
>
> *Reply:*
>
> We apologize for the lack of clarity in explaining the connection between sufficient representation and ensemble learning. We have revised Section 5.1 of the manuscript to provide a more detailed explanation of the concept behind ensemble learning.

---

> ### Author Response · Authors · 2024-11-28
> **Gentle Reminder: Further Discussion**
>
> Dear reviewer ZUgG,
>
> We really appreciate your acknowledgment of our contribution, which makes us feel it is our obligation to improve the quality of this paper. We are looking forward to hearing from your further opinions about our rebuttal. We cherish the opportunity to polish our paper through this interactive discussion and hope you can spare a little time to take a look.
>
> We would be glad to have further discussions if there are any suggestions or questions. Thank you again for your effort and valuable comments.
>
> Best regards,
>
> The Authors.

---

> > ### Comment · Reviewer_ZUgG · 2024-12-01
> >
> > Thanks for your response. However, most of concerns remain unresolved, especially regarding the usefulness of sufficient representation condition and the number of baselines (since no further experiments are added). Therefore I decide to keep my score.

---

### Official Review · Reviewer_vFLM · 2024-11-05

**Soundness:** 2
**Presentation:** 2
**Contribution:** 2
**Rating:** 3
**Confidence:** 4

**Summary:**

This submission considers domain generalization, where the learner observes data from source environments/domains and needs to learn a hypothesis that generalizes well to unseen environments. A particular structural causal model is assumed, where there are core and spurious latent factors, such that only the core latent factor causally influences the label and does it with an invariant $P(Y | Z)$ distribution. To make the problem well-defined, the authors make two assumptions: (i) a label identifiability condition similar to $P(Y|X)$ being invariant and (ii) that every possible value of the core variable is supported in at least one of the training domains. Furthermore, the authors assume infinite data from each domain, but finite training domains.

Under the stated assumptions, the authors present two necessary and two sufficient conditions for obtaining a hypothesis that is optimal for all domains, including the unseen ones.
* Necessary condition #1 (See definition 3.3) is simple and states that a globally optimal (i.e., optimal for all possible domains) hypothesis should be optimal for training domains too.
* Sufficient condition #1 (Theorem 3.4) states that if there are sufficiently many diverse training domains, then any hypothesis that is optimal for all training domains is also globally optimal.
* Sufficient condition #2 (Theorem 3.6) states that hypotheses that consist of an optimal (w.r.t. training domains) classification head on top of an invariant representation ($P(Y|Z)$ being invariant for *all* domains) are globally optimal.
* Necessary condition #2 (Theorem 3.8) states that globally optimal hypotheses that consist of a classification head on top of a representation $z = g(x)$ should be such that the representation function $g$ can be reduced to an invariant representation, i.e. there exists a function $\psi$ such that $P(Y | \psi(g(X)))$ is invariant.

The authors also show that the two necessary conditions together are not enough to guarantee global optimality. Based on the above results, the authors recommend designing domain generalization algorithms that strive for the sufficient conditions but making sure that the necessary conditions are satisfied. They propose a new domain generalization algorithm that partitions the input space and enforces similar marginal distributions of representations across environments conditioned in each part.

**Strengths:**

* Understanding necessary and/or sufficient conditions for domain generalization is an important problem and can lead to effective domain generalization algorithms. The authors formulate a reasonable structural causal model and assumptions that enable formal analysis.
* Stressing the importance of having sufficient representations (Definition 3.7 and necessary condition #2) and pointing out that many domain generalizations fail to find a globally optimal hypothesis partly for not having sufficient representations.

**Weaknesses:**

**Clarity and quality of writing.** The paper is hard to read and is unclear in many parts. Below are a few places where the writing can be improved.
* Lines 49-50: “In other approaches grounded in causality, there is typically an assumption of arbitrary targets” – this is unclear.
* Lines 050-053: At this point of the text it is hard to understand these two sentences. Writing can also be improved.
* Lines 076-083: It is unclear what sufficient conditions are referred to here.
* Equation (2) should be before equation (1) for better readability.
* Theorem 3.4 should state clearly where x comes from (the support of $P^{e_k}$ as I understand).
* Lines 195-196: Optimality for training domains is not really verifiable, given that we have only finite data from the training environment in practice.
( Lines 438-442: Unclear what is multi-view representation here. The notation does not make sense here.

**Not enough rigor in mathematical expositions.** I found some of the proofs hard to follow and possibly flawed.
* I am not convinced of the proof of Theorem 3.4. In fact, the Theorem 3.4 might not hold, because while with each added “diverse” environment at least one function is eliminated, this does not mean that in the limit only globally optimal functions would be left. Since a general function space is typically uncountable, the process might not converge as one adds countably infinite training domains.
* The paper is very loose about $\ell(y’, y)$ and $y’=f(x)$. Sometimes $y’$ is a distribution over classes, but sometimes it is a predicted hard label.
* The proof of Corollary 4.1 is unclear, although I believe the result is correct.

**Weak evidence that the proposed algorithm outperforms the baselines.**
* Looking at the result tables, the improvements over baselines seems marginal and can possible be attributed to the proposed method having more hyperparameters (number of subspaces per class, $\lambda_P$, and $\lambda_D$).
* As shown in Table 3, varying the number of subspaces does not have a large effect. To better motivate and understand the proposed method, it would be good to design a synthetic 2D domain generalization task and visualize what subspaces are found.
* It would be interesting to add a baseline consisting of only SRA to separate its effect from that of SWAD.

**Questions:**

**A few minor comments**
* Proposition 3.5 needs a fix, it should be $P(y|g(x))$.
* Theorem A.5 is wrongly stated in the appendix, it repeats Theorem A.6.
* Results tables presented in Appendix have many cases where the bolded value is actually smaller than another value in the same column.
* Theorem 3.6 requires one domain to cover the whole $\mathcal{Z}_c$, which is not exactly what Assumption 3.2 is.
In my opinion this work needs a focal point. Currently it is unclear what the main takeaways and results are.

---

> ### Author Response · Authors · 2024-11-23
> **Responses for Weakness 1**
>
> We sincerely appreciate your thorough and insightful feedback. Below, we outline our efforts to address the valuable points you have raised.
>
> &nbsp;
>
> **Weakness 1: Clarity and quality of writing**
>
> &nbsp;
>
>  **Lines 49-50:** *“In other approaches grounded in causality, there is typically an assumption of arbitrary targets” – this is unclear.*
>
>   We have revised the manuscript to: *"In other approaches grounded in causality, there is typically an assumption of having prior knowledge of target domains."*
>
> &nbsp;
>
> **Lines 50-53:** *At this point of the text, it is hard to understand these two sentences. Writing can also be improved.*
>
>   We have revised these sentences for clarity.
>
> &nbsp;
>
> **Lines 76-83:** *It is unclear what sufficient conditions are referred to here.*
>
>   We have revised the manuscript for this paragraph to clearly reference the mentioned sufficient and necessary conditions.
>
>
> &nbsp;
>
> **Equation (2) should be before Equation (1) for better readability.**
>
>  Following the reviewer's suggestion, we have revised the manuscript and moved Equation (2) before Equation (1) for better readability.
>
> Additionally, to enhance clarity, we rigorously define the loss function:
>   $\ell:\mathcal{Y}\_{\Delta} \times \mathcal{Y} \rightarrow \mathbb{R}$
>   where $\ell\left(f(x), y\right)$ with $f(x) \in \mathcal{Y}_{\Delta}$ and $y \in \mathcal{Y}$ specifies the loss (e.g., cross-entropy) for assigning a data sample $x$ to the class $y$ using the hypothesis $f$.
>
> &nbsp;
>
> **Theorem 3.4 should state clearly where $x$ comes from the support of $\mathbb{P}^{e_k}$, as I understand$.**
>
>   Following the reviewer's suggestion, we have revised Theorem 3.4 to clarify the origin of \(x\).
>
> &nbsp;
>
> **Lines 195-196:** *Optimality for training domains is not really verifiable, given that we have only finite data from the training environment in practice.*
>
> While our work considers limited and finite domains, we follow recent theoretical works (Wang et al., 2022a; Rosenfeld et al., 2020; Kamath et al., 2021; Ahuja et al., 2021; Chen et al., 2022b) that assume an infinite data setting for each training environment. This assumption differentiates domain generalization (DG) literature from traditional generalization analysis (e.g., PAC-Bayes framework), which focuses on in-distribution generalization where testing data are drawn from the same distribution. We explicitly state this presumption in Section 2.2, "Domain Generalization Setting."
>
> &nbsp;
>
> **Lines 438-442:** *Unclear what is multi-view representation here. The notation does not make sense here.*
>
>
> We apologize for the lack of clarity in the notation. The term "multi-view representation" refers specifically to an ensemble of representations. In particular,
>
> We resort to maximizing the lower bound $I(Z; Y)$ to increasing the chance of learning $Z$ that contains causal information $Z_c$.
>
> Recall that we use the cross-entropy loss $\ell:\mathcal{Y}_{\Delta} \times \mathcal{Y} \mapsto \mathbb{R}$ to optimize the hypothesis for training domains. It is well-known that minimizing the cross-entropy loss is equivalent to maximizing the lower bound of $I(Z; Y)$  (Qin et al., 2019; Colombo et al., 2021). In other words, hypotheses that are optimal on training domains (Condition 3.3) also promote the sufficient representation function condition (Condition 3.7).
>
> However, maximizing the lower bound $I(Z; Y)$ only ensures that $Z$ captures the shared information $I(X; Y)$ and potentially some additional information about $Z_c$ (as illustrated in Figure 3 (Right)).
>
> To encourage the representation $Z$ to capture more information from $Z_c$, this approach can be extended to learn multiple versions of representations through ensemble learning. Specifically, we can learn an $M$-ensemble of representations $Z^M$:
>
> $$Z^M = \left\\{Z\_i = g\_i(X) \mid g\_i \in \arg\max\_{g\_i} I(g\_i(X); Y) \right\\}\_{i=1}^{M},$$
>
> to capture as much information as possible about $Z_c$.
>
> We have revised the manuscript and updated the notation from "multi-view representation" to "ensemble of representations" for improved clarity.

---

> ### Author Response · Authors · 2024-11-23
> **Responses for Weakness 2**
>
> **Weakness 2: Not enough rigor in mathematical expositions**
>
> &nbsp;
>
> **I am not convinced of the proof of Theorem 3.4.** * In fact, the Theorem 3.4 might not hold, because while with each added “diverse” environment at least one function is eliminated, this does not mean that in the limit only globally optimal functions would be left. Since a general function space is typically uncountable, the process might not converge as one adds countably infinite training domains.*
>
> *Reply:*
>
> Since our SCM is restricted to the space $\mathcal{E}$, and the sequence of training domains is given by $\mathcal{E}_{tr}=\\{e_1,...,e_K\\} \subseteq \mathcal{E}$, we propose to modify
>
>
> $\lim\_{K\rightarrow \infty}\mathcal{F}\_{\cap}^{K} \rightarrow \mathcal{F}^*.$
>
> to
>
> $\lim\_{\mathcal{E}\_{tr}\rightarrow \mathcal{E}}\mathcal{F}\_{\cap}^{ \left| \mathcal{E}_{tr} \right|} \rightarrow \mathcal{F}^*.$
>
> We hope this addresses the reviewer's concern, and we look forward to further discussion.
>
> &nbsp;
>
> **The paper is very loose about $\ell(y',y)$ and $y'=f(x)$ . Sometimes $y'$ is a distribution over classes, but sometimes it is a predicted hard label.**
>
> *Reply:*
>
> We apologize for the ambiguity in the definition of \(\ell\). In the revised version, we have redefined the loss function as follows:
>
>   $\ell:\mathcal{Y}\_{\Delta} \times \mathcal{Y} \rightarrow \mathbb{R}$
>   where $\ell\left(f(x), y\right)$ with $f(x) \in \mathcal{Y}_{\Delta}$ and $y \in \mathcal{Y}$ specifies the loss (e.g., cross-entropy) for assigning a data sample $x$ to the class $y$ using the hypothesis $f$.
>
> and updated all related discussions and proofs to reflect the new definition of $\ell$.
>
> &nbsp;
>
> **The proof of Corollary 4.1 is unclear, although I believe the result is correct.**
>
> *Reply:*
>
> We apologize for the lack of clarity. In the revised version, we have updated the proof of Corollary 4.1.

---

> ### Author Response · Authors · 2024-11-23
> **Responses for Weakness 3**
>
> **Weakness 3: Weak evidence that the proposed algorithm outperforms the baselines.**
>
> &nbsp;
>
> **It would be interesting to add a baseline consisting of only SRA to separate its effect from that of SWAD.**
>
> *Reply:*
>
> We have included a baseline consisting solely of SRA without SWAD. While SRA achieves the best average performance on the benchmark without SWAD, it does not consistently outperform across all datasets compared to the settings with SWAD.
>
> Additionally, we have included further experimental results in Section C.4, comparing our method with other baselines from DomainBed.
>
> For the baselines with SWAD, due to time constraints, we have not yet completed the experiments on the DomainNet dataset. We will update the results as soon as they are available.
>
> &nbsp;
>
> **Looking at the result tables, the improvements over baselines seems marginal and can possible be attributed to the proposed method having more hyperparameters (number of subspaces per class, $\lambda_P$, and $\lambda_D$).**
>
> *Reply:*
>
> We would like to clarify the relationship between SRA, DANN, and CDANN. Specifically, SRA (ours), DANN, and CDANN all utilize $\mathcal{H}$-divergence for alignment, and all three methods share the parameter $\lambda_D$.
>
> The key difference lies in what they align:
> - DANN aligns the entire domain representation.
> - CDANN aligns class-conditional representations.
> - SRA aligns subspace-conditional representations (e.g., $\left| \mathcal{M}\right|$ subspaces per class; additionally, the hyper-parameter $\lambda_P$ determines how the subspaces are organized).
>
> DANN and CDANN can be considered specific cases of our approach. In particular, CDANN is equivalent to SRA with $\left| \mathcal{M}\right| = 1$.
>
>
>
> Additionally, on average, SRA outperforms DANN/CDANN by approximately 2% without SWAD and 1.8% with ensemble learning, which represents a significant improvement in the DG benchmark. Furthermore, as shown in the additional results in Section C.4, the baselines fail to consistently surpass the simple ERM baseline across all settings. While some methods perform well on certain datasets, they perform worse on others. In contrast, SRA combined with SWAD consistently outperforms all baselines across all settings.
>
> Finally, It is worth noting that we use the same values for $\left| \mathcal{M} \right|$, $\lambda_P$, and $\lambda_D$ across all experiments.
>
> &nbsp;
>
> **As shown in Table 3, varying the number of subspaces does not have a large effect. To better motivate and understand the proposed method, it would be good to design a synthetic 2D domain generalization task and visualize what subspaces are found.**
>
> *Reply:*
>
> As shown in Table 5 in Section C.4, the performance gap between the baselines on the PACS dataset is relatively small. Consequently, the 0.6% difference between the worst-case $\left|\mathcal{M}\right| = 4$ and the best-case $\left|\mathcal{M}\right| = 16$ is not negligible. Furthermore, when compared to CDANN $\left|\mathcal{M}\right| = 1$), the gap becomes more significant, reaching 1%.
>
> The optimal number of subspaces is indeed related to the number of causal factors, which may vary depending on the dataset. Analyzing this relationship becomes challenging with high-dimensional data. We are currently working on designing a synthetic 2D domain generalization benchmark and hope to provide updates in the revised version as soon as possible.

---

> ### Author Response · Authors · 2024-11-23
> **Responses for Questions**
>
> **A few minor comments**
>
> We apologize for the errors in presentation. In the revised version, we have updated Proposition 3.5, Theorem A.5, the results tables presented in the Appendix, and other unclear sections highlighted by the reviewers.
>
> &nbsp;
>
> **Theorem 3.6** requires one domain to cover the whole , which is not exactly what Assumption 3.2 is. In my opinion this work needs a focal point. Currently it is unclear what the main takeaways and results are.
>
> For simplicity, we present Theorem 3.6 with a general domain $\mathbb{P}^e\$. Assumption 3.2 ensures that we can learn the optimal hypothesis given knowledge of the invariant representation function \(g_c\) from the training domains. Specifically, we consider a mixture of training domains with the following support:
>  $\cup_{e\in \mathcal{E}_{tr}}\text{supp}\{\mathbb{P}^{e} \left (Z_c \right )\}=\mathcal{Z}_c$.
>
> &nbsp;
>
> **main takeaways**
>
> 1. The DG literature often overlooks the importance of sufficient representation function (Condition 3.7) in real-world scenarios. This oversight occurs because algorithms that satisfy sufficient conditions are assumed to automatically fulfill the sufficient representation function (Condition 3.7). However, achieving sufficient conditions is almost impossible in finite-domain settings. DG analysis, which often focuses on idealized settings where sufficient conditions are more likely to be satisfied, can lead to misleading assumptions about Condition 3.7 in practical applications.
> For instance, methods like IIB (Li et al., 2022a) and IB-IRM (Ahuja et al., 2021), which incorporate the information bottleneck principle, may inadvertently harm the sufficient representation function (Condition 3.7).
>
> 2. We recommend designing domain generalization (DG) algorithms that strive for the sufficient conditions while ensuring that the necessary conditions are also met.

---

> ### Author Response · Authors · 2024-11-28
> **Gentle Reminder: Further Discussion**
>
> Dear reviewer vFLM,
>
> We really appreciate your acknowledgment of our contribution, which makes us feel it is our obligation to improve the quality of this paper. We are looking forward to hearing from your further opinions about our rebuttal. We cherish the opportunity to polish our paper through this interactive discussion and hope you can spare a little time to take a look.
>
> We would be glad to have further discussions if there are any suggestions or questions. Thank you again for your effort and valuable comments.
>
> Best regards,
>
> The Authors.

---

### Official Review · Reviewer_uuZ4 · 2024-11-06

**Soundness:** 2
**Presentation:** 2
**Contribution:** 2
**Rating:** 5
**Confidence:** 3

**Summary:**

This paper addresses the domain generalization (DG) problem by exploring it through the lens of sufficient and necessary conditions for generalization. The authors demonstrate that sufficient conditions are often unverifiable in practice. They then introduce two necessary conditions to ensure the performance of DG methods. Using this framework, they analyze and explain existing DG approaches. Finally, they propose a new method called Subspace Representation Alignment (SRA) and demonstrate its effectiveness across various datasets.

**Strengths:**

1. The paper studies the domain generalization problem from the perspective of necessary and sufficient conditions, which is interesting and provides a fresh viewpoint.
2. The theoretical contributions are generally sound and contribute to a deeper understanding of DG.
3. The Subspace Representation Alignment (SRA) method proposed by the authors achieves good performance on various datasets.

**Weaknesses:**

1. There are several concerns regarding Theorem 5.1:
   - It appears that the authors have omitted a $1/|\mathcal{E}_{tr}|$ term in the first term on the right-hand side (RHS) of equation (i) in Theorem 1.
   - The first term on the RHS is actually the same as the left-hand side (LHS), as verified by Equation (17) in the appendix. As a result, the inequality in Theorem 5.1 seems useless.
   - Theorem A.11, which seems to be a restatement of Theorem 5.1, differs significantly from Theorem 5.1. It is unclear which version is correct and should be included in the main text.
2. The connection between Theorem 5.1 and the proposed method requires further elaboration. It appears that the authors aim to use Equation (4) to optimize the upper bound of the LHS of condition (i) in Theorem 5.1. However, it seems feasible to optimize the LHS directly, raising questions about the necessity of introducing the subspace representation alignment term in Equation (4).
3. The two necessary conditions proposed in the paper seem to align with the objectives of Invariant Risk Minimization (IRM). A more detailed comparison with IRM would strengthen the paper and clarify the contributions.
4. There are issues with notation and definitions:
   - The use of $h \in \arg\min \sup$ in Equation (4) is unusual. In the definition of $\mathcal{F}^*$, a predictor $f \in \mathcal{F}^*$ if and only if for all $e$, $f \in \arg\min L(f, P^e)$. However, Equation (4) presents a minimax problem, which differs from the definition of $\mathcal{F}^*$. Should it be $\forall e, h \in \arg\min L(h \circ g, P^e)$ in Equation (3)?
5. There are unclear parts and typos:
   - The sentence is incomplete in Line 51 and needs revision
   - The notation $\Phi$ in Definition 3.7 is introduced without explanation, causing confusion.
   - $G_c$ should be $\mathcal{G}_c$ in Line 240.

**Questions:**

See the weakness part.

---

> ### Author Response · Authors · 2024-11-23
> **Responses for Weakness 1 and  Weakness 2**
>
> We sincerely appreciate your thorough and insightful feedback. Below, we outline our efforts to address the valuable points you have raised.
>
> &nbsp;
>
> **Weakness 1: There are several concerns regarding Theorem 5.1**
>
> &nbsp;
>
> **Q1.** It appears that the authors have omitted a $\frac{1}{\left |\mathcal{E}_{r} \right |}$ term in the first term on the right-hand side (RHS) of equation (i) in Theorem 1.
>
> *Reply:*
>
> It should be $\left | \mathcal{E}\_{r}\right |$ instead of $\frac{1}{\left| \mathcal{E}\_{r} \right |}$ on the left-hand side (LHS). We apologize for this error. The manuscript has been updated in both the main text and the appendix, and the proof of this theorem (A11) is now provided in greater detail (highlighted in blue).
>
> &nbsp;
>
> **Q2:** The first term on the RHS is actually the same as the left-hand side (LHS), as verified by Equation (17) in the appendix. As a result, the inequality in Theorem 5.1 seems useless.
>
> *Reply:*
>
> We would like to recall the theoretical result presented in Theorem 4.2:
>
> $
> D\left(\mathbb{P}\_{\mathcal{Y}}^{e},\mathbb{P}\_{\mathcal{Y}}^{e'}\right) \leq D\left ( g\_{\\#}\mathbb{P}^{e},g\_{\\#}\mathbb{P}^{e'} \right )+\mathcal{L}\left ( f,\mathbb{P}^{e} \right )+\mathcal{L}\left ( f,\mathbb{P}^{e'} \right )
> $
>
> This result suggests that if there is a substantial discrepancy $D\left(\mathbb{P}\_{\mathcal{Y}}^{e},\mathbb{P}\_\{\mathcal{Y}}^{e'}\right)$ in the label marginal distribution across training domains, enforcing *representation alignment* by minimizing $D\left( g\_{\\#}\mathbb{P}^{e}, g\_{\\#}\mathbb{P}^{e'} \right)$ will increase *domain losses* $\mathcal{L}\left( f, \mathbb{P}^{e} \right) + \mathcal{L}\left( f, \mathbb{P}^{e'} \right).$. (as the LHS > 0, the RHS cannot be optimized to zero)
>
> &nbsp;
>
> Referring back to Theorem 5.1 (i), optimizing the LHS results in the baseline ERM. However, when applying domain generalization (DG) algorithms, particularly representation-based techniques like DANN or CDANN (optimizing RHS), a trade-off emerges between achieving alignment in representations and minimizing the training domain loss (optimal hypothesis for training domains condition).
>
> In Theorem 5.1 (ii), we demonstrate that, with appropriately organized subspaces, it is possible to simultaneously optimize both representation alignment and training domain loss (optimal hypothesis for training domains condition) without any trade-offs.
>
> &nbsp;
>
> **Q3:** Theorem A.11, which seems to be a restatement of Theorem 5.1, differs significantly from Theorem 5.1. It is unclear which version is correct and should be included in the main text.
>
> *Reply:*
>
> The trade-off described in Theorem 4.2 has been analyzed theoretically in several works, including Phung et al. (2021) with Hellinger distance, Zhao et al. (2019) with $\mathcal{H}$-divergence, and Le et al. (2021) with Wasserstein distance.
>
> In Theorems 4.2 and 5.1 of the main paper, we present the general framework for three divergences/distances: $\mathcal{H}$-divergence, Wasserstein distance, and Hellinger distance. In the Appendix, we provide a detailed theoretical proof for Hellinger distance, and a similar approach can be directly applied to $\mathcal{H}$-divergence and Wasserstein distance.
>
> It is worth noting that this trade-off was first analyzed by Zhao et al. (2019). However, to the best of our knowledge, we are the first to propose an approach to address it.

---

> ### Author Response · Authors · 2024-11-23
> **Responses for Weakness 3**
>
> **Weakness 3:** The two necessary conditions proposed in the paper seem to align with the objectives of Invariant Risk Minimization (IRM). A more detailed comparison with IRM would strengthen the paper and clarify the contributions.
>
> *Reply:*
>
> We have presented additional experimental results in Section C.4, comparing our method to other baselines, including IRM and other common baselines from DomainBed.
>
> The reason we focus on DANN and CDANN is that they are the two most closely related methods, as they represent specific cases of our approach. Specifically, SRA (ours), DANN, and CDANN all utilize $\mathcal{H}$-divergence for alignment.
>
> The key difference lies in what they align:
> - DANN aligns the entire domain representation.
> - CDANN aligns class-conditional representations.
> - SRA align subspace-conditional representations (e.g., $\left| \mathcal{M}\right|$ subspaces per class).
>
> CDANN is equivalent to SRA with $\left| \mathcal{M}\right| = 1$.
>
>
> In detail, as presented in Theorems 4.2 and 5.1, DANN and CDANN suffer from a trade-off between optimizing alignment and minimizing training domain losses, while SRA addresses this issue effectively.
>
> At a higher level, DANN and CDANN aim to achieve invariant representations (a sufficient condition) but impose problematic constraints, as the trade-off with training domain losses can harm the necessary conditions for generalization. In contrast, SRA imposes well-structured constraints, allowing it to simultaneously encourage alignment and minimize training domain losses.
> This validates our theoretical conclusions: enforcing good sufficient conditions (as in SRA) while promoting necessary conditions can significantly improve generalization.

---

> ### Author Response · Authors · 2024-11-23
> **Responses for Weakness 4 and Weakness 5**
>
> **Weakness 4:** There are issues with notation and definitions:
>
> The use of $h\in \arg \min \sup$ in Equation (4) is unusual. In the definition of $\mathcal{F}^*$, a predictor $f\in \mathcal{F}^*$ if and only if for all $e$, $\underset{ f}{\rm{argmin }}\mathcal{L}(f,\mathbb{P}^e)$. However, Equation (4) presents a minimax problem, which differs from the definition of $\mathcal{F}^*$. Should it be $\forall e$ $h\in \underset{h}{\rm{argmin }}\mathcal{L}(h\circ g,\mathbb{P}^e)$
>  in Equation (3)?
>
> *Reply:*
>
> In Equation (3) we define
>
> $\mathcal{F}^{B}\_{\mathbb{P}^e,g}=\left \\{ h\circ g \mid h \in \underset{ h'\in \mathcal{H}\_{\mathbb{P}^e,g}}{\rm{argmin }} \sup\_{e'\in \mathcal{E}} \mathcal{L}\left (  h'\circ g, \mathbb{P}^{e'} \right ) \right \\}$
>
> We apologize for the ambiguity caused by this definition.
>
> Our idea is to demonstrate that if $g$ is not a sufficient representation ($g \notin \mathcal{G}_s$), it is impossible to find any $h$ such that $h \circ g \in \mathcal{F}^*$. Equation (3) focuses on verifying whether the optimal classifier $h$ for the worst-case domain still belongs to the set of optimal hypotheses \(\mathcal{F}^*\). However, this definition seems unclear and may cause confusion.
>
> Following the reviewer’s suggestion, we have updated it to:
>
> $\mathcal{F}^{B}\_{\mathbb{P}^e, g}= \left \\{ h \circ g \mid h \in \bigcap\_{e^{'} \in \mathcal{E}} \underset{ h^{'}\in\mathcal{H\}\_{\mathbb{P}^e,g}}{\rm{argmin }}\mathcal{L}\left (  h^{'}\circ g, \mathbb{P}^{e^{'}} \right )  \right \\}$
>
>
>
> We thank the reviewer once again for valuable correction.
>
> &nbsp;
>
> **Weakness 5:** There are unclear parts and typos.
>
> *Reply:*
>
> We apologize for the unclear parts and typos in the manuscript. We have revised the manuscript in accordance with the reviewer's comments.

---

> > ### Comment · Reviewer_uuZ4 · 2024-11-27
> >
> > I thank the authors for their response. However, after reading the other reviews, I remain negative about this paper and believe that a major revision is needed, particularly to correct the errors and make the theoretical results more rigorous. As a result, I am keeping my score unchanged.

---

> > > ### Author Response · Authors · 2024-11-27
> > >
> > > Thank you for your response. We have made every effort to address the concerns raised by the reviewer and believe we have also addressed the concerns of other reviewers. At this stage, we feel it would not be reasonable to adjust our work further based on other reviewers' comments.
> > >
> > > Once again, we sincerely thank the reviewer for taking the time to review our paper. Your feedback has been invaluable in helping us improve the quality of this work.

---

### Meta-Review · Area_Chair_WmTN · 2024-12-18

**Metareview:**

Summary: This paper studies the sufficient and necessary conditions for the domain generalization problem. The paper shows that sufficient conditions are oftentimes unverifiable in practice. Therefore, the authors turn to necessary conditions to analyze existing DG approaches. They also propose an algorithm, Subspace Representation Alignment, to learn sufficient invariant representations.

Strengths:
1. Necessary and sufficient conditions of domain generalization problems are important research topic. The paper provides a fresh perspective on this topic.

Weaknesses:
1. As raised by Reviewer uuZ4, there are errors in Theorem 5.1 (the lack of $|\epsilon_r|$ term), and the authors admit it in the rebuttal. Errors are not acceptable in the top conferences such as ICLR.
2. Presentation errors exist in Proposition 3.5, Theorem A.5, the results tables presented in the Appendix, and other unclear sections highlighted by the reviewers.
3. One reviewers have concerns on the readability of the proof. The proof may have errors and be non-rigorous.
4. The paper shows weak evidence that the proposed algorithm outperforms the baselines.

All reviewers vote for rejection consistently. AC would follow reviewers' suggestions and recommend for rejection as well. AC would encourage the authors to take the above weaknesses into consideration in the future revision of this paper.

**Additional Comments On Reviewer Discussion:**

In the first round of review, all reviewers vote for rejection consistently. After the rebuttal phase, the reviewers find the rebuttal non-convincing and most of concerns remain unresolved. So they prefer to keep their original scores. AC would follow reviewers' suggestions and recommend for rejection as well.

---

### Decision · Program_Chairs · 2025-01-22

Reject